# Efficiently Controlling Multiple Risks with Pareto Testing

**Bracha Laufer-Goldshtein, Adam Fisch, Regina Barzilay & Tommi Jaakkola**
CSAIL, MIT, {`lauferb,fisch,regina,tommi`}`@csail.mit.edu`

## Abstract

Machine learning applications frequently come with multiple diverse objectives and constraints that can change over time. Accordingly, trained models can be tuned with sets of hyper-parameters that affect their predictive behavior (e.g., their run-time efficiency versus error rate). As the number of constraints and hyper-parameter dimensions grow, naively selected settings may lead to sub-optimal and/or unreliable results. We develop an efficient method for calibrating models such that their predictions provably satisfy multiple explicit and simultaneous statistical guarantees (e.g., upper-bounded error rates), while also optimizing any number of additional, unconstrained objectives (e.g., total run-time cost). Building on recent results in distribution-free, finite-sample risk control for general losses, we propose Pareto Testing: a two-stage process which combines multi-objective optimization with multiple hypothesis testing. The optimization stage constructs a set of promising combinations on the Pareto frontier. We then apply statistical testing to this frontier only to identify configurations that have (i) high utility with respect to our objectives, and (ii) guaranteed risk levels with respect to our constraints, with specifiably high probability. We demonstrate the effectiveness of our approach to reliably accelerate the execution of large-scale Transformer models in natural language processing (NLP) applications. In particular, we show how Pareto Testing can be used to dynamically configure multiple inter-dependent model attributes—including the number of layers computed before exiting, number of attention heads pruned, or number of text tokens considered—to simultaneously control and optimize various accuracy and cost metrics.

## 1 Introduction

Suppose you want to deploy a modern machine learning model in a real-world environment. As a practitioner, you may frequently have to weigh several performance considerations (Jin & Sendhoff, 2008; Ribeiro et al., 2020; Min et al., 2021). For example, how much computational budget can you spend? What accuracy do you require? How large, if any, of a discrepancy in predictive performance across different groups of end-users can you tolerate? Often models are equipped with hyper-parameter configurations that provide "knobs" for tuning different aspects of their performance, depending on how such questions are answered. As the number of parameter dimensions and objectives grow, however, choosing the right set of parameters to rigorously control model performance on test data in the intended ways can become prone to error.

To address this challenge, the recently proposed *Learn Then Test* (LTT) framework of Angelopoulos et al. (2021) combines any type of parameterizable predictive model with classic statistical hypothesis testing to provide an algorithm for selecting configurations that lead to provable distribution-free, finite-sample risk control of any user-specified objective. Nevertheless, while theoretically general, a key pair of practical challenges arises when the space of parameters to explore and constraints to satisfy are large. The first is that evaluating all possible configurations can quickly become intractable, while the second is that the statistical tests relied upon to guarantee risk control can quickly lose power—and fail to identify configurations that are also useful for the task at hand.

In this work, we build upon the results of LTT by introducing *Pareto Testing*, a simple procedure that can provide a computationally and statistically efficient way to identify valid, risk-controlling configurations with (specifiably) high probability, which, critically, are also useful with respect to other objectives of interest. Our method consists of two stages. In the first stage, we solve an unconstrained,

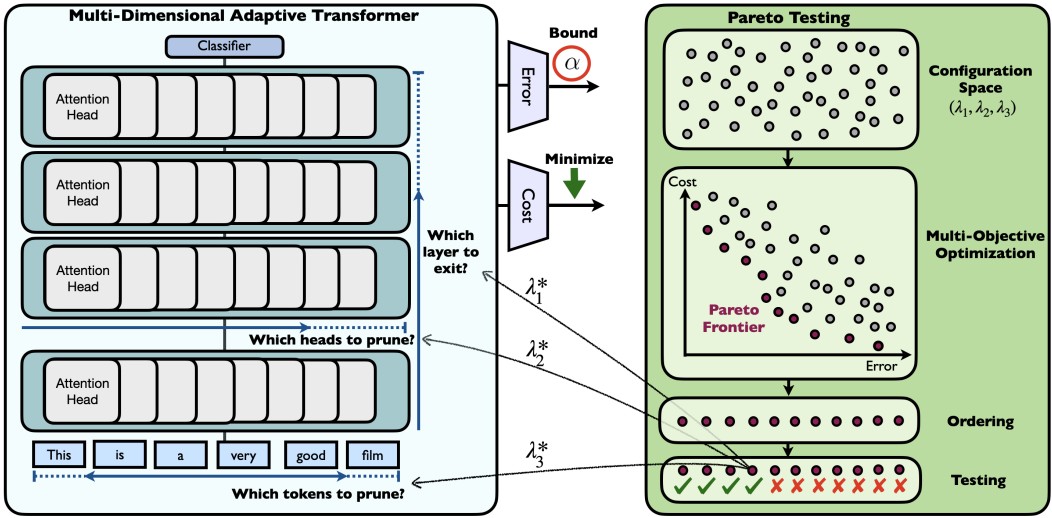

Figure 1: A demonstration of our calibration procedure applied to multi-dimensional adaptive computation in a Transformer model (left). Here we have the option to drop tokens from the input, make an "early-exit" prediction after processing a subset of the layers, or only compute a subset of the self-attention heads in each layer in order to do faster inference. Our calibration procedure (right) applies multi-objective optimization to identify a Pareto frontier of configurations with different performance profiles, and then applies statistical testing to efficiently identify a subset of "risk-controlling" configurations with high probability (e.g., bounded error rates).

multi-objective optimization problem in order to recover an approximate set of Pareto-optimal configurations, i.e., settings for which no other configuration exists that is uniformly better in all respects. Here we can exploit standard multi-objective optimization methods to efficiently explore and filter large parameter spaces to only its most promising configurations. In the second stage, we perform rigorous sequential testing over the recovered set, which we empirically find to yield tight control of our desired risks, while also giving good performance with respect to our free objectives.[1]

We apply our approach to adaptive computation in large-scale Transformer models (Vaswani et al., 2017) for natural language processing (NLP), see Figure 1. While larger models generally perform better, they can also be incredibly computationally intensive to run (Bapna et al., 2020; Schwartz et al., 2020; Moosavi et al., 2021). Often, however, not every application, domain, or example requires the same amount of computation to achieve similar performance. As such, many techniques have been proposed for accelerating computation, including attention head pruning, token dropping, or early exiting (Graves, 2016; Xin et al., 2020; Hou et al., 2020; Goyal et al., 2020). Still, determining the extent to which to apply different modifications while still preserving good performance can be tricky. Our proposed procedure allows the user to jointly configure *multiple* model settings subject to *multiple* statistical guarantees on model performance—such as average and worst-case relative reductions in accuracy (e.g., so that the adaptive model is within 5% of the full model's accuracy), average inference cost (e.g., so that the adaptive model uses less than a certain number of FLOPS on average), or maximum abstention rates in selective prediction settings.

**Contribution.** The core idea and contribution of our work can be summarized quite plainly:

1. Our framework leverages statistical testing techniques via the LTT framework (Angelopoulos et al., 2021) to identify valid risk-controlling hyper-parameter configurations;
2. To improve efficiency, we introduce Pareto Testing, our main contribution, as a way to efficiently guide the number and order of configurations that we test when searching for valid settings;
3. We demonstrate the scalability and effectiveness of our method in managing trade-offs in multi-dimensional adaptive computation in NLP applications with large-scale Transformer models;
4. On diverse text classification tasks, we empirically achieve tight, simultaneous control of multiple risks while also improving performance on any non-controlled objectives, relative to baselines.

---

[1]If we fail to find any valid configurations (which may not exist) with the right confidence, then we abstain.

## 2 RELATED WORK

**Risk control.** Our work adds to a rich history of tools for uncertainty estimation and risk control for machine learning algorithms (Vovk, 2002; Vovk et al., 2015; 2017; Lei et al., 2013; 2018; Gupta et al., 2020; Bates et al., 2021; Barber et al., 2021; Angelopoulos et al., 2021). Here we focus on achieving model-agnostic, distribution-free, and finite-sample performance guarantees—similar to the coverage guarantees given by prediction sets or regression intervals in conformal prediction (Papadopoulos et al., 2002; Vovk et al., 2005; Angelopoulos et al., 2022). As outlined in §1, this paper builds on the groundwork set by Angelopoulos et al. (2021), which provides a general methodology for calibrating any risk function that is controllable via some low-dimensional hyper-parameter configuration. We extend their framework to efficiently handle (relatively) higher-dimensional settings with multiple auxiliary objectives. Our application to confident model acceleration is also closely related to Schuster et al. (2021; 2022), though our method is designed for a much broader setting that involves (i) multiple objectives, and (ii) multiple model pruning dimensions.

**Multi-objective optimization.** Solving for multiple objectives is a fundamental problem (Deb, 2001; Miettinen, 2012; Bradford et al., 2018). Typically, multi-objective problems are more difficult than single-objective problems, as a single solution does not always exist due to trade-offs. Instead, there is a set of solutions that are all equally "good", which is known as the Pareto frontier (Censor, 1977; Arora, 2004). Our setting falls at the intersection of multi-objective optimization and risk-control, where we want to perform multi-objective optimization subject to statistical bounds on a subset of the objectives. Our two-stage approach is able to directly combine techniques in multi-objective optimization (Knowles, 2006; Lindauer et al., 2022) with those in risk control (Angelopoulos et al., 2021), in order to identify valid, statistically efficient solutions.

**Model configuration.** We approach our multi-objective optimization problem by uncovering model configurations that deliver on the desired performance guarantees (e.g., bounded error rates), while also providing "best-effort" optimization of the auxiliary objectives (e.g., minimal inference cost) without any re-training. This is adjacent to the field of hyper-parameter tuning and architecture search, which deals with determining appropriate model hyper-parameter values, or even designing higher-level network structures (Elsken et al., 2019). While most approaches focus on finding configurations that maximize predictive performance, some have also considered additional measures such as efficiency (Shah & Ghahramani, 2016; Belakaria et al., 2019; Elsken et al., 2018; Dong et al., 2018; Zhou et al., 2018; Chu et al., 2020), fairness (Schmucker et al., 2020; Candelieri et al., 2022), or robustness (Karl et al., 2022). Our work, however, differs by treating hyper-parameter selection as a multiple-testing problem with rigorous statistical guarantees following Angelopoulos et al. (2021).

**Adaptive computation.** Our main application is configuring adaptive model computation. Large-scale deep learning models can be accurate, but also computationally intensive to run. Many efforts have been focused on improving run-time efficiency, including model distillation (Sanh et al., 2019; Jiao et al., 2020; Sun et al., 2020), dynamic architecture selection (Yu et al., 2019; Cai et al., 2020; Hou et al., 2020), early-exiting (Teerapittayanon et al., 2016; Liu et al., 2020), token pruning (Goyal et al., 2020; Ye et al., 2021; Kim et al., 2021; Modarressi et al., 2022; Guan et al., 2022), and others. This work focuses on configurable, adaptive computation that does not require model re-training. Furthermore, only a few methods have proposed *combining* multiple pruning dimensions, such as depth and width (Hou et al., 2020), token pruning with early-exiting (He et al., 2021), and pruning model units of different granularity (Xia et al., 2022). Our multi-dimensional calibration scheme generalizes these approaches, and allows for flexible tuning in each pruning axis.

## 3 PROBLEM FORMULATION

Consider an input variable $X \in \mathcal{X}$, and an associated label $Y \in \mathcal{Y}$, drawn from some joint distribution. Assume a predictive model of the form $f \colon \mathcal{X} \times \Lambda \to \mathcal{Y}$, where $\Lambda \triangleq \Lambda_1 \times \ldots \times \Lambda_n$ is the space of $n$ hyper-parameters $(\lambda_1, \ldots, \lambda_n)$ that configure $f$. The parameters of the model $f$ are optimized over a training set $\mathcal{D}_{\text{train}}$, while the hyper-parameters then provide $n$ additional degrees of freedom in influencing either (i) how the model is trained over $\mathcal{D}_{\text{train}}$, or (ii) how the model is used. We focus on the latter in this paper. For example, in our adaptive Transformer example, $f$ has $n = 3$ pruning dimensions: the number of attention heads per layer, the truncated length of the input text sequence, and the effective network depth in terms of the selected early-exit layer. In this

scenario, the hyper-parameters $(\lambda_1, \lambda_2, \lambda_3)$ are real-valued thresholds, and determine the extent of sparsification along each axis given some "importance/confidence" score (to be defined later).

Next, consider a set of objective functions $\{Q_1, \ldots, Q_{c+k}\}$ of the form $Q_i(\lambda_1, \ldots, \lambda_n) = \mathbb{E}[q_i(X, Y; \lambda_1, \ldots, \lambda_n)]$ for some loss function $q_i$. Under our setting, we assume that the user wishes to arbitrarily bound the first $c$ objective functions (hereby called *risk* functions) by $\{\alpha_1, \ldots, \alpha_c\} \in \mathbb{R}^c$, while also minimizing the remaining $k$ objective functions. We further assume that $(\lambda_1, \ldots, \lambda_n)$ can be used to either increase or decrease each objective $Q_i$ (not necessarily independently), although we do *not* assume that all values of $Q_i$ are jointly achievable.

To estimate suitable hyper-parameter values, let $\mathcal{D}_{\text{cal}} = (X_i, Y_i)$, $i = 1, \ldots, m$ be an i.i.d. *calibration* set that we will use to select $(\hat{\lambda}_1, \ldots, \hat{\lambda}_n)$. As functions of $\mathcal{D}_{\text{cal}}$, $(\hat{\lambda}_1, \ldots, \hat{\lambda}_n)$ are also random variables, and therefore $Q_i(\hat{\lambda}_1, \ldots, \hat{\lambda}_n)$ is a *random* conditional expectation (constant given $\mathcal{D}_{\text{cal}}$). Our goal is to select $(\hat{\lambda}_1, \ldots, \hat{\lambda}_n)$ in such a way that the $\{Q_1, \ldots, Q_c\}$ objectives that we wish to control are appropriately bounded with specifiably high probability—as we now formally define.

**Definition 3.1** (($\alpha, \delta$)-risk controlling configuration). *Let $\mathcal{D}_{\text{cal}} = \{(X_i, Y_i)\}_{i=1}^m$ be i.i.d. random variables that are used to estimate a model configuration $(\hat{\lambda}_1, \ldots, \hat{\lambda}_n)$. Let $\{Q_i(\hat{\lambda}_1, \ldots, \hat{\lambda}_n)\}_{i=1}^c$ be a set of risk functions conditioned on the choice of $(\hat{\lambda}_1, \ldots, \hat{\lambda}_n)$. For any set of risk levels $\{\alpha_i\}_{i=1}^c$ and tolerance $\delta \in (0, 1)$, we say that $(\hat{\lambda}_1, \ldots, \hat{\lambda}_n)$ is a $(\alpha, \delta)$-risk controlling configuration, if:[2]*

$$\mathbb{P}\left(Q_i(\hat{\lambda}_1, \ldots, \hat{\lambda}_n) \leq \alpha_i\right) \geq 1 - \delta, \ \ \forall i \in \{1, \ldots, c\} \quad simultaneously, \tag{1}$$

*where the probability in Eq. (1) is over the draw of $\mathcal{D}_{\text{cal}}$.*

Many satisfactory configurations may exist for a given task and constraints. Our key practical goal is to find a $(\alpha, \delta)$-risk controlling configuration that *also* best minimizes the remaining objectives, $\{Q_{c+1}, \ldots, Q_{c+k}\}$. As such, we focus on the expected performance of $Q_{c+j}(\hat{\lambda}_1, \ldots, \hat{\lambda}_n)$, for $j \in \{1, \ldots, k\}$ as a relative measure of effectiveness when comparing selected configurations.[3]

## 4 BACKGROUND

We briefly review the Learn then Test framework of Angelopoulos et al. (2021), multiple hypothesis testing, and multiple objective optimization—which comprise the key components of our method.

**Learn then Test (LTT).** The core idea of LTT is to use multiple hypothesis testing to rigorously select a risk controlling configuration $\boldsymbol{\lambda} = (\lambda_1, \ldots, \lambda_n) \in \Lambda$. Consider a single risk $Q$. A set of possible configurations $\Lambda_g$ is chosen for testing, usually by defining a discrete grid over the configuration space $\Lambda$. For each configuration $\boldsymbol{\lambda} \in \Lambda_g$, LTT tests the null hypothesis, $H_{\boldsymbol{\lambda}}: Q(\boldsymbol{\lambda}) > \alpha$, i.e., that the risk is *not* controlled. A successful rejection of $H_{\boldsymbol{\lambda}}$ then implies that $\boldsymbol{\lambda}$ *is* risk-controlling.

A valid (super-uniform) p-value to use as a basis for accepting or rejecting $H_{\boldsymbol{\lambda}}$ can be derived from concentration bounds on the empirical risk. For example, Hoeffding's inequality can be used to yield

$$p^{\text{cal}}(\boldsymbol{\lambda}, \alpha) = p(\hat{Q}^{\text{cal}}(\boldsymbol{\lambda}); \alpha, m) = e^{-2m\left(\alpha - \hat{Q}^{\text{cal}}(\boldsymbol{\lambda})\right)_+^2}, \tag{2}$$

so that $\mathbb{P}(p^{\text{cal}}(\boldsymbol{\lambda}, \alpha) \leq u) \leq u$, for all $u \in [0, 1]$. Similar p-values can be derived from different bounds. Here, we will use the more powerful Hoeffding-Bentkus p-value throughout (Appendix A). A subset of valid $\boldsymbol{\lambda}$ configurations is then selected out of $\Lambda_g$ by applying a family-wise error rate (FWER) controlling procedure, so that the probability of making one or more false discoveries (i.e., choosing *invalid* $\boldsymbol{\lambda}$) is bounded by $\delta \in (0, 1)$. This can be extended to *multiple* risk control by defining the combined null hypothesis, $H_{\boldsymbol{\lambda}} : \exists\, i$ where $Q_i(\boldsymbol{\lambda}) > \alpha_i$. A valid combined p-value can be obtained by taking the maximum p-value over all objective functions (see Appendix A for a proof):

$$p^{\text{cal}}(\boldsymbol{\lambda}, \boldsymbol{\alpha}) = \max_{1 \leq i \leq c} p(\hat{Q}_i^{\text{cal}}(\boldsymbol{\lambda}); \alpha_i, m), \quad \text{with } \boldsymbol{\alpha} = (\alpha_1, \ldots, \alpha_c). \tag{3}$$

**Multiple hypothesis testing.** A key component of LTT is the choice FWER-controlling procedure. As the number of tested hypotheses $H_{\boldsymbol{\lambda}}$ grows (e.g., for combinatorially many $\boldsymbol{\lambda}$), the harder it is to

---

[2]This is a slight abuse of terminology in that, technically, $(\hat{\lambda}_1, \ldots, \hat{\lambda}_n)$ as a random variable is not necessarily a configuration that achieves risk control, but rather its realizations are valid with the appropriate probability.

[3]This is analogous to using the average set size to compare conformal predictors (Vovk et al., 2005; 2016).

reject $H_{\boldsymbol{\lambda}}$ while limiting the probability of false discoveries to $\delta$. Different FWER-controlling procedures have different statistical efficiency/power (i.e., ability to correctly reject $H_{\boldsymbol{\lambda}}$ when it is false). Angelopoulos et al. (2021) consider a number of FWER-controlling procedures, namely the Bonferroni correction, Fixed Sequence Testing (FST), and Sequential Graphical Testing (SGT)—see Appendix E for a complete discussion. At a high level, the Bonferroni correction assigns an "error budget" of $\delta/|\Lambda_g|$ to each possible $\boldsymbol{\lambda} \in \Lambda_g$. For large $|\Lambda_g|$, this strict tolerance can result in conservative rejections. FST and SGT attempt to exploit *structure* in $\Lambda$ by ordering and testing $\boldsymbol{\lambda} \in \Lambda_g$ in ways that are likely to result in more valid rejections. FST considers a sequence of hypothesis tests in some order, and terminates when the first null hypothesis $H_{\boldsymbol{\lambda}}$ fails to be rejected. The efficiency of FST relies heavily on the ordering of hypothesis tests (e.g., those that are more likely to be rejected should be tested earlier). Defining a proper ordering for FST is challenging for a large, possibly unstructured (or with unknown structure) $\Lambda$. This challenge intensifies when combined with the *additional* goal of finding configurations that not only provide multiple risk control, but also optimize $Q_{c+1}, \ldots, Q_{c+k}$.

**Multiple objective optimization.** Generally speaking, multi-objective optimization minimizes a vector-valued function $\mathbf{g}(\boldsymbol{\lambda}) : \Lambda \to \mathbb{R}^r$, where $\mathbf{g}(\boldsymbol{\lambda}) = [G_1(\boldsymbol{\lambda}), \ldots, G_r(\boldsymbol{\lambda})]^T$ consists of $r$ objectives $G_i$ (we use $G$ to avoid confusion with $Q$ for now). For nontrivial multi-objective function $\mathbf{g}(\boldsymbol{\lambda})$ with conflicting objectives (e.g., accuracy vs. cost), there is no single point that minimizes all objectives simultaneously. Instead, for any $\boldsymbol{\lambda}, \boldsymbol{\lambda}' \in \Lambda$, we say that $\boldsymbol{\lambda}'$ *dominates* $\boldsymbol{\lambda}$ ($\boldsymbol{\lambda}' \prec \boldsymbol{\lambda}$), if for every $i \in \{1, \ldots r\}$, $G_i(\boldsymbol{\lambda}') \leq G_i(\boldsymbol{\lambda})$, and for some $i \in \{1, \ldots r\}$, $G_i(\boldsymbol{\lambda}') < G_i(\boldsymbol{\lambda})$. In other words, $\boldsymbol{\lambda}'$ dominates $\boldsymbol{\lambda}$ if there is no objective for which $\boldsymbol{\lambda}$ is superior to $\boldsymbol{\lambda}'$, and for at least one objective $\boldsymbol{\lambda}'$ is strictly better. The *Pareto optimal set* consists of all points that are not dominated by any point in $\Lambda$:

$$\Lambda_{\text{par}} = \{\boldsymbol{\lambda} \in \Lambda : \ \{\boldsymbol{\lambda}' \in \Lambda : \ \boldsymbol{\lambda}' \prec \boldsymbol{\lambda}, \boldsymbol{\lambda}' \neq \boldsymbol{\lambda} \ \} = \emptyset\}. \tag{4}$$

## 5 Pareto testing

We now present our method for selecting effective risk-controlling configurations. We adopt the strategy of Split FST (Angelopoulos et al., 2021) for separating the calibration data into two disjoint subsets $\mathcal{D}_{\text{opt}}$ and $\mathcal{D}_{\text{testing}}$ of sizes $m_1$ and $m_2$, respectively. The first split is used for defining an ordered sequence of configurations to test, while the second is used to conduct the hypothesis tests.

### 5.1 Constructing the Pareto frontier

We begin with defining a set of tests/configurations to consider. We solve the multi-objective optimization problem defined by the vector-valued function $\mathbf{q}(\boldsymbol{\lambda}) : \Lambda \to \mathbb{R}^{c+k}$ that consists of all of the objective functions (both constrained and unconstrained), i.e. $\mathbf{q}(\boldsymbol{\lambda}) = [Q_1(\boldsymbol{\lambda}), \ldots, Q_{c+k}(\boldsymbol{\lambda})]^T$. Practically, we use $\mathbf{q}^{\text{opt}}(\boldsymbol{\lambda})$ with empirical objectives $\hat{Q}_i^{\text{opt}}(\boldsymbol{\lambda})$ defined over $\mathcal{D}_{\text{opt}}$. Any efficient solver for multi-objective optimization can be used to approximate the Pareto optimal set $\Lambda_{\text{par}}$. We will show results with both brute-force optimization over a grid of configurations, as well as with a multi-objective Bayesian optimizer (Lindauer et al., 2022).

The main idea of our method is to perform testing only over the Pareto optimal set. This consists of the most "promising" configurations, and provides the best achievable trade-offs between all objectives (with respect to $\mathcal{D}_{\text{opt}}$). As mentioned earlier, a major challenge when dealing with a large hyper-parameter space is that testing numerous configurations can quickly lead to a loss in statistical efficiency. We overcome this by focusing only on the "optimal" region of the hyper-parameter space.

### 5.2 Ordering the Pareto frontier

We now define an ordering for the set of tests/configurations on the Pareto frontier along which to conduct FST. We take the simple, but empirically effective, strategy of ordering $\boldsymbol{\lambda} \in \Lambda_{\text{par}}$ by their (combined) estimated p-values $p^{\text{opt}}(\boldsymbol{\lambda}, \boldsymbol{\alpha}) = \max_{1 \leq i \leq c} p(\hat{Q}_i^{\text{opt}}(\boldsymbol{\lambda}); \alpha_i, m_1)$, which we compute over $\mathcal{D}_{\text{opt}}$ (the same data used to recover the Pareto frontier, but separate from testing data). Converting the $c$ constrained dimensions to p-values and taking their maximum, allows us to align and compare risks of different types (e.g., binary 0/1 error vs. precision/recall rates in $[0, 1]$), or that are controlled by different bounds (e.g., $\alpha_i \ll \alpha_j$). Intuitively, because we focus on the Pareto optimal set, for each configuration along this ordering there is no other configuration with a lower estimated p-value that is also expected to be dominant on the $k$ free objectives. Note that for $c > 1$, we can

---

**Algorithm 1** Pareto Testing

---

**Definitions:** $f$ is a configurable model with $n$ thresholds $\boldsymbol{\lambda} = (\lambda_1, \ldots, \lambda_n)$. $\mathcal{D}_{\text{cal}} = \mathcal{D}_{\text{opt}} \cup \mathcal{D}_{\text{testing}}$ is a calibration set of size $m$, split into optimization and (statistical) testing sets of size $m_1$ and $m_2$, respectively. $\{Q_1, \ldots, Q_{c+k}\}$ are objective functions. $\boldsymbol{\alpha} = \{\alpha_1, \ldots, \alpha_c\}$ are user-specified risk bounds for the first $c$ objectives. $\Lambda$ is the configuration space. $\delta$ is the tolerance. PARETOOPTIMALSET returns the Pareto frontier, and can either be computed via a multi-objective optimization algorithm or exhaustive search (see Algorithm F.1).

1: **function** OPTIMIZATION($\mathcal{D}_{\text{opt}}, \Lambda, \boldsymbol{\alpha}$)
2: $\quad \Lambda_{\text{par}} \leftarrow$ PARETOOPTIMALSET($\Lambda, \hat{Q}_1^{\text{opt}}, \ldots, \hat{Q}_{c+k}^{\text{opt}}$)
3: $\quad p^{\text{opt}}(\boldsymbol{\lambda}, \boldsymbol{\alpha}) \leftarrow \max_{1 \le i \le c} p(\hat{Q}_i^{\text{opt}}(\boldsymbol{\lambda}); \alpha_i, m_1)$, for all $\boldsymbol{\lambda} \in \Lambda_{\text{par}}$
4: $\quad \Lambda_{\text{ordered}} \leftarrow$ Order configurations according to increasing $p^{\text{opt}}(\boldsymbol{\lambda}, \boldsymbol{\alpha})$
5: $\quad$ **return** $\Lambda_{\text{ordered}}$

6: **function** CALIBRATION($\mathcal{D}_{\text{testing}}, \Lambda_{\text{ordered}}, \boldsymbol{\alpha}, \delta$)
7: $\quad \hat{Q}_i^{\text{testing}}(\boldsymbol{\lambda}) \leftarrow \frac{1}{m_2} \sum_{(X,Y) \in \mathcal{D}_{\text{testing}}} q_i(X, Y; \boldsymbol{\lambda})$, for all $\boldsymbol{\lambda} \in \Lambda_{\text{ordered}}$, and $1 \le i \le c$
8: $\quad p^{\text{testing}}(\boldsymbol{\lambda}, \boldsymbol{\alpha}) \leftarrow \max_{1 \le i \le c} p(\hat{Q}_i^{\text{testing}}(\boldsymbol{\lambda}); \alpha_i, m_2)$, for all $\boldsymbol{\lambda} \in \Lambda_{\text{ordered}}$
9: $\quad$ Apply FST: $\Lambda_r = \{\boldsymbol{\lambda}^{(j)} : j < J\}$, $J = \min_j \{j : p^{\text{testing}}(\boldsymbol{\lambda}^{(j)}, \boldsymbol{\alpha}) \ge \delta\}$
10: $\quad \Lambda^* \leftarrow$ PARETOOPTIMALSET($\Lambda_r, \hat{Q}_{c+1}^{\text{testing}}, \ldots, \hat{Q}_{c+k}^{\text{testing}}$)
11: $\quad$ **return** $\Lambda^*$

---

(optionally) prune the frontier by considering only the subset $\Lambda'_{\text{par}} \subseteq \Lambda_{\text{par}}$ that is optimal with respect to $\tilde{\mathbf{q}}^{\text{opt}}(\boldsymbol{\lambda}, \boldsymbol{\alpha}) = [p^{\text{opt}}(\boldsymbol{\lambda}, \boldsymbol{\alpha}), \hat{Q}_{c+1}^{\text{opt}}(\boldsymbol{\lambda}), \ldots, \hat{Q}_{c+k}^{\text{opt}}(\boldsymbol{\lambda})]^T$. In other words, since we only care about the *maximum* p-value over the constrained objectives, we can ignore configurations in $\Lambda_{\text{par}}$ that only differ along the constrained dimensions, without affecting the free dimensions or the combined p-value.

## 5.3 APPLYING FIXED SEQUENTIAL TESTING ON THE PARETO FRONTIER

After defining and ordering $\Lambda_{\text{par}}$ over $\mathcal{D}_{\text{opt}}$, we then proceed with FST over $\mathcal{D}_{\text{testing}}$ to identify a subset $\Lambda_r \subseteq \Lambda_{\text{par}}$ of configurations for which we can successfully reject $H_{\boldsymbol{\lambda}}$ (i.e., that are valid risk-controlling configurations). Finally, after obtaining the validated subset $\Lambda_r$, we again find and return the resulting Pareto frontier (now, with respect to only the objectives of practical interest: $\{Q_{c+1}, \ldots, Q_{c+k}\}$). For $k = 1$, this consists of a single configuration $\boldsymbol{\lambda}^*$, while for $k > 1$, this consists of a set of non-dominated, valid configurations $\Lambda^*$.

Our method is summarized in Algorithm 1 and is illustrated in Figure 2. We call it *Pareto Testing* for two reasons: (i) the method reduces to applying FST over a path defined by the Pareto front of the multi-objective problem, and (ii) repeated testing for different $\boldsymbol{\alpha}$ limitations yields a *calibrated* Pareto frontier with constraints on specific dimensions in the objective function space. It is straightforward to show that Pareto Testing achieves valid risk control, as we now formally state.

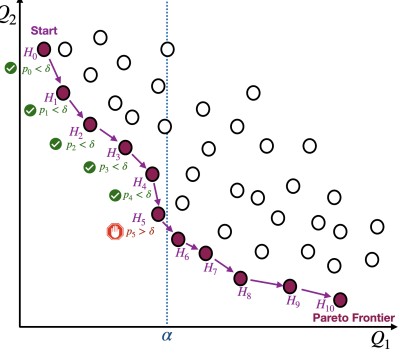

Figure 2: Pareto Testing with two objectives. $Q_1$ is controlled at $\alpha$ while $Q_2$ is minimized. FST is applied along the sequence of configurations on the Pareto front, ordered from low to high expected risk w.r.t. $Q_1$.

**Proposition 5.1.** *Let $\mathcal{D}_{\text{cal}} = \{(X_i, Y_i)\}_{i=1}^m$ be a set of i.i.d. random variables split into two disjoint subsets, $\mathcal{D}_{\text{opt}}$ and $\mathcal{D}_{\text{testing}}$. Let $p^{\text{testing}}(\boldsymbol{\lambda}, \boldsymbol{\alpha})$ be a valid p-value for a configuration $\boldsymbol{\lambda}$, where $\mathbb{P}(p^{\text{testing}}(\boldsymbol{\lambda}, \boldsymbol{\alpha}) \le u) \le u$ for all $u \in [0, 1]$ over the draw of $\mathcal{D}_{\text{testing}}$. Then all configurations in the output set $\Lambda^*$ of Algorithm 1 are also simultaneously $(\alpha, \delta)$-risk controlling configurations.*

The proof, given in Appendix A, follows from Split FST. Note that for $k > 1$, the chosen set $\Lambda^*$ contains configurations that are simultaneously valid. We are therefore free to use any $\boldsymbol{\lambda}_i^* \in \Lambda^*$, as well as any randomly *combined* configuration in the convex hull of $\Lambda^*$ defined as follows. Consider a randomized strategy, where for each test point, we sample the configuration $\boldsymbol{\lambda}_j^* \in \Lambda^*$ to use with probability $\Delta_j$, where $\Delta$ lies in the $|\Lambda^*| - 1$ dimensional probability simplex. The resulting combination is also $(\alpha, \delta)$-risk controlling, and allows for different (average) outcomes on the $k$ free objectives.

**Corollary 5.2.** *Any randomized time-shared use of configurations in $\Lambda^*$ is $(\alpha, \delta)$-risk controlling.*

## 6  ADAPTIVE MULTI-DIMENSIONAL PRUNING

We now turn to a concrete application of our method, in which we demonstrate its effectiveness for reliably accelerating Transformer models (Vaswani et al., 2017). Here we pair each pruning "dimension" with a score function that estimates the relative importance of each prunable element in that dimension. By thresholding scores, we obtain different versions of the model with different performance. We assume a $K$-layer Transformer model with $W$ attention heads per layer, and $L(X)$ input tokens (see Vaswani et al. (2017) for a complete description of the Transformer model).

### 6.1  CONFIGURABLE DIMENSIONS

We consider the following adaptive behaviors (see also Appendix B for details):

1. **Token pruning.** We assign each token with an (after-the-fact) importance score, based on the gradient of the output probability w.r.t. the token embedding. To determine token importance at run-time, we predict the score at each layer (Modarressi et al., 2022), then remove tokens with estimated scores bellow a threshold $\lambda^{\text{tok}}$, yielding a sequence of size $L_j(X; \lambda^{\text{tok}})$ at the $j$-th layer.
2. **Early exiting.** We attach a softmax classifier head to each layer, and exit whenever the predictive entropy of its output is below a threshold $\lambda^{\text{layer}}$. The exit layer is denoted as $K_{\text{exit}}(X; \lambda^{\text{layer}})$.
3. **Head pruning.** Similar to token pruning, we compute an importance score for each attention head per layer by the gradient of the loss over the validation set (from training) w.r.t the head hidden representation, following Michel et al. (2019). $W_j(\lambda^{\text{head}})$ denotes the number of retained heads in layer $j$. Note that this is fixed for all inputs, unlike the previous mechanisms.

### 6.2  OBJECTIVE FUNCTIONS

We also define several practical objective functions. $f(\cdot; \lambda_0)$ denotes the full model without pruning.

**Relative computational cost.** We define the relative computations cost by the ratio between the computational cost of the pruned model and the computational cost of the full model:

$$Q_{\text{cost}}(\lambda) = \mathbb{E}\left[ \frac{\sum_{j=1}^{K_{\text{exit}}(X;\lambda^{\text{layer}})} W_j(\lambda^{\text{head}}) \cdot L_j(X; \lambda^{\text{tok}})^2}{\sum_{j=1}^{K} W \cdot L(X)^2} \right]. \tag{5}$$

Eq. (5) reflects a simplistic definition that incorporates a quadratic dependency on the sequence length due to the attention mechanism, and a linear dependency on the number of attention heads. We also consider total FLOPs (floating-point operations) per forward pass.

**Relative accuracy reduction.** Speeding up the run-time of a model can also degrade its accuracy. Define the random variable $D(X, Y; \lambda) = \mathbf{1}\{f(X; \lambda_0) = Y\} - \mathbf{1}\{f(X; \lambda) = Y\}$, which is 0 when both model predictions are the same, 1 when the full model is correct while the pruned model is incorrect, and -1 if the opposite is true. We define the relative accuracy reduction as:

$$Q_{\text{acc}}(\lambda) = \mathbb{E}\left[D(X, Y; \lambda)\right] = \mathbb{E}\left[\mathbf{1}\{f(X; \lambda_0) = Y\}\right] - \mathbb{E}\left[\mathbf{1}\{f(X; \lambda) = Y\}\right], \tag{6}$$

i.e, the difference in accuracy between the full and pruned models. In order to exploit p-values derived from confidence bounds that assume the risk is in $[0, 1]$ (Angelopoulos et al., 2021), we define $D'(X, Y; \lambda) = [D(X, Y; \lambda)]_+$, which differs only for the rare event that the pruned model is correct while the full model is not, and is more restrictive since $\mathbb{E}[D(X, Y; \lambda)] \leq \mathbb{E}[D'(X, Y; \lambda)]$.

**Worst-class relative accuracy reduction.** In some cases, we would like to control the worst-class accuracy, or equivalently, that every class accuracy reduction is controlled by the same level:

$$Q_{\text{acc-class}}(y; \lambda) = \mathbb{E}\left[D'(X, Y; \lambda) \mid Y = y\right] \leq \alpha, \ \ \forall y \in \mathcal{Y}. \tag{7}$$

Note that this adds an additional $|\mathcal{Y}|$ objectives (that can still be solved efficiently, see Appendix A).

**Selective classification abstention rate.** Consider a selective classification problem, where the model is allowed to abstain from making a prediction when it is unsure, based on some threshold $\tau$ on the model's confidence (we use the probability of the predicted class $\max_y f(X, y; \lambda)$). In this case, we re-define the relative accuracy and cost reductions to be conditioned on making a prediction. We also introduce abstention rate objective (e.g., abstain from prediction at most $20\%$ of the time):

$$Q_{\text{abstention-rate}}(\lambda, \tau) = \mathbb{E}\left[\mathbf{1}\{\max_y f(X, y; \lambda) < \tau\}\right]. \tag{8}$$

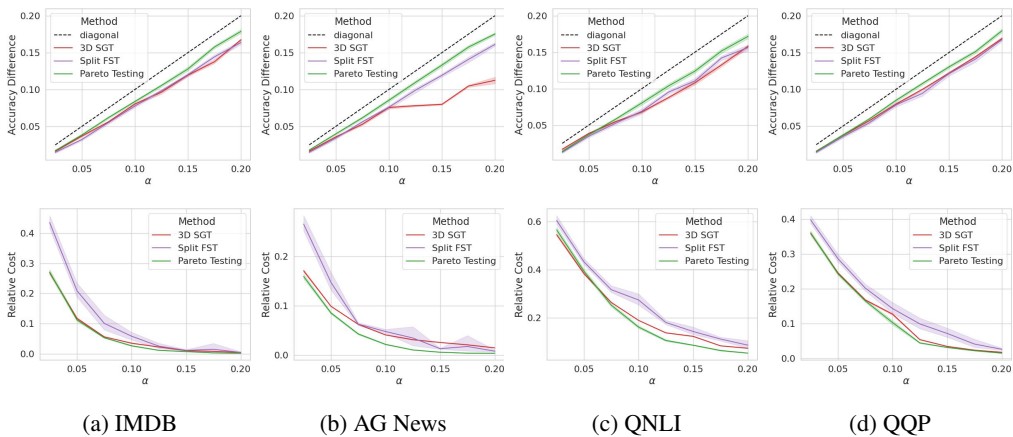

(a) IMDB        (b) AG News        (c) QNLI        (d) QQP

Figure 3: Two objectives (100 random splits). Relative accuracy reduction is controlled, while computational cost is minimized. Top plots show accuracy differences; bottom plots show relative cost.

## 7 EXPERIMENTS

**Experimental setup.** we test our method over five text classification tasks of varied difficulty levels: **IMDB** (Maas et al., 2011), **AG News** (Zhang et al., 2015), **QNLI** (Rajpurkar et al., 2016), **QQP**, **MNLI** (Williams et al., 2018). We use a BERT-base model (Devlin et al., 2018) with $K = 12$ layers and $W = 12$ heads per layer, and attach a prediction head and a token importance predictor per layer.

**Baselines and Evaluation.** We present both risk controlling and non-risk-controlling baselines. Non-risk-controlling baselines: (1) $\alpha$-**constrained**, the solution to the constrained optimization problem in Eq. (21); (2) $(\alpha, \delta)$-**constrained**, the same as before, but with constraints defined over p-values, which is equivalent to testing without FWER control. Risk-controlling baselines: (3) **3D SGT**, SGT defined over a 3D graph, see Algorithm F.3; (4) **Split FST**, the split method proposed in Angelopoulos et al. (2021), where a set of hypotheses is ordered by increasing estimated p-values. For fairness, each baseline (including our method) operates over the same predefined grid of configurations. We use $6480$ configurations in total (18 head, 20 token, and 18 early-exit thresholds). Note that the recovered Pareto front in this case is restricted to points in this grid, see Algorithm F.1. We also show the results obtained while using a multi-objective optimizer to demonstrate the actual *computationally* efficient implementation of our method (rather than brute-force exploration of the grid). We repeat each experiment over different splits of calibration and test data (50-100 runs in total), and report the mean over all splits (with $95\%$ CIs) for the configurations selected by each method.

**Two objectives (one controlled, one free).** We start with a two-objective scenario, where we wish to control the accuracy reduction (Eq. (6)), while minimizing the cost (Eq. (5)). The average accuracy reduction and relative cost are presented in Fig. 3 for the risk controlling baselines. We observe that the proposed method obtains the lowest cost among the risk controlling baselines for all $\alpha$ values and across all tasks. In particular, it can be seen that Split FST obtains slightly looser control of relative accuracy reduction, but higher relative computational costs compared to Pareto Testing. Ordering by p-values alone does not take into account scenarios where several configurations have similar accuracy, but vary in costs, while the proposed method optimizes the selection and ordering of configurations in both dimensions. We also see that 3D-SGT performs well for low $\alpha$ values, but often becomes worse as $\alpha$ increases. A possible factor is that as $\alpha$ increases, 3D testing is allowed to explore more of the 3D graph, but does so inefficiently—leading to overall lower rejection rates. Figure 4 shows the difference between the risk controlling and the non-risk-controlling baselines in terms of satisfying Definition 3.1. In non-risk controlling baselines (left), the risk exceeds $\alpha$ more frequently than the allowed tolerance level $\delta = 0.1$. By contrast and as expected, we see that all the risk controlling baselines (right) are always below the tolerance level.

**Three objectives (two controlled, one free).** We study a scenario with three objectives on MNLI, where we control both the average accuracy (Eq. (6)) and the worst-class accuracy (Eq. (7)) while minimizing cost (Eq. (5)). We vary the values of $\alpha_1$ for average accuracy and set $\alpha_2 = 0.15$ for worst accuracy. Figure 5 reports the results of the three objective functions. It can be seen that when

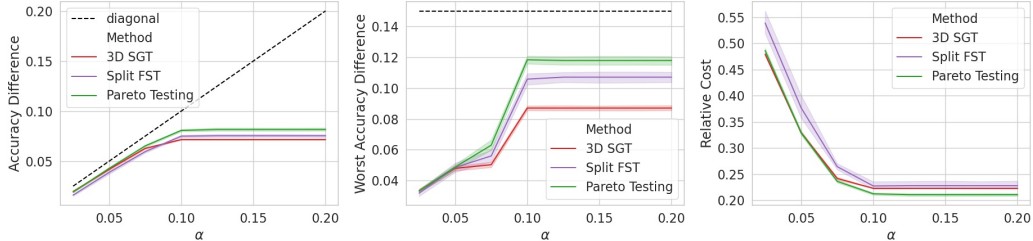

Figure 4: Two-objectives, QNLI (100 splits). Acc. reduction is controlled, cost is minimized. Left: histogram of acc. reduction, $\alpha = 0.05$; middle: violin plots of acc. reduction; right: risk violations.

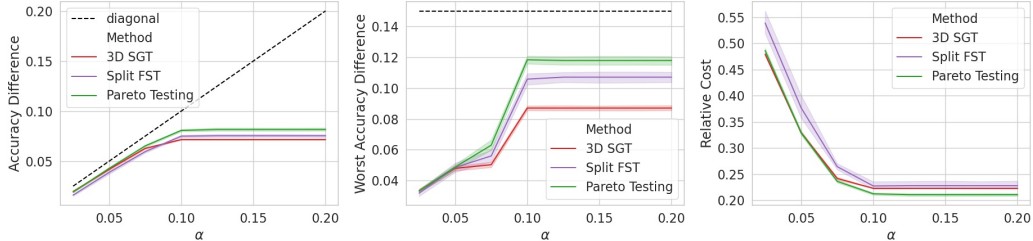

Figure 5: Three-objectives, MNLI (100 random splits): average accuracy is controlled by $\alpha_1 \in \{0.025, 0.5, \ldots, 0.2\}$, worst accuracy is controlled by $\alpha_2 = 0.15$, $\delta = 0.1$, and cost is minimized.

$\alpha_1$ is small, testing is dominated by average accuracy (worst accuracy is not tight), and as $\alpha_1$ increases, worst accuracy becomes dominant and average accuracy becomes loosely controlled. Here too we see that Pareto Testing obtains improved cost reduction with respect to the other baselines.

**Results with an off-the-shelf optimizer.** On the first scenario with accuracy control and cost minimization, in Figure D.1 we show a comparison between the proposed method with grid (blue) and multi-objective optimizer (red to yellow) with different number of function evaluations. For multi-objective optimizer, we used an implementation (Lindauer et al., 2022) of ParEGO optimization algorithm (Knowles, 2006; Cristescu & Knowles, 2015). We observe that even with a small number of evaluations (e.g., 50), we obtain reasonable results, which further improve as the number of evaluations increases. The grid option performs better for certain $\alpha$ values, but it requires significantly more function evaluations. A more in depth analysis of how the multi-objective optimization method and the allowed number of evaluations influence testing efficiency is left for future work.

**Additional results.** We briefly highlight a number of results contained in Appendix D. On the same "accuracy control" setting, we report FLOPs saved, as an alternative measure for cost improvement. In addition, we show flipped results for controlling cost while minimizing the relative loss in accuracy. We also explore a selective prediction setting with three objectives when one is controlled while two are free. Specifically, we control the selective accuracy loss (Eq. (17)), while minimizing both the selective cost (Eq. (16)) and the abstention rate (Eq. (8)). Figure D.2 reports the cost and abstention rate for the chosen configurations by either the proposed method and Split FST. It can be seen that Pareto Testing selects a richer set of configurations that conveys better cost-coverage trade-offs.

## 8    CONCLUSION

Deployment of machine learning models in the real world can frequently demand precise guarantees that certain constraints will be satisfied, together with good empirical performance on other objectives of interest. In this work, we presented *Pareto Testing*, a two-stage procedure for multiple risk control combined with multi-objective optimization. In the first stage, Pareto Testing relaxes all constraints, and converts the problem to a standard multi-objective optimization format that can be efficiently solved with off-the-shelf optimizers to yield a Pareto frontier of hyper-parameter configurations affecting model performance. In the second stage, this Pareto frontier is filtered via multiple hypothesis testing to identify configurations that simultaneously satisfy the desired risk constraints with (specifiably) high probability—while also being effective solutions for the free objectives. Bridging theory and practice, we demonstrated the effectiveness of our method for reliable and efficient adaptive computation of Transformer models on several text classification tasks under various conditions.

## ACKNOWLEDGMENTS

We thank the anonymous reviewers and members of the Barzilay and Jaakkola research groups for helpful discussions and feedback. B.L.G. is supported in part by the CHE Data Science fellowship, the Zuckerman-CHE STEM Leadership Program, the Schmidt Futures Israeli Women's Postdoctoral Award and the Technion Viterbi Fellowship. A.F. is supported in part by the NSF Graduate Research Fellowship. This work is also supported in part by the ML for Pharmaceutical Discovery and Synthesis (MLPDS) Consortium and the DARPA Accelerated Molecular Discovery program.

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

## A    MATHEMATICAL DETAILS

We present the proofs for our theoretical claims.

### A.1    MAX P-VALUE FOR MULTIPLE RISKS

First we re-state and re-prove that taking the maximum p-value is also a valid p-value.

**Lemma A.1.** *Let $p_i(\boldsymbol{\lambda}, \boldsymbol{\alpha})$ be a p-value for $H_{\boldsymbol{\lambda},i} : Q_i(\boldsymbol{\lambda}) > \alpha_i$, for each $i \in \{1, \ldots, c\}$. Define $p(\boldsymbol{\lambda}, \boldsymbol{\alpha}) := \max_{1 \le i \le c} p_i(\boldsymbol{\lambda}, \alpha_i)$. Then, for all $\boldsymbol{\lambda}$ such that $H_{\boldsymbol{\lambda}} : \exists i$ where $Q_i(\boldsymbol{\lambda}) > \alpha_i$ holds, we have:*

$$\mathbb{P}\left(p(\boldsymbol{\lambda}, \boldsymbol{\alpha}) \le u\right) \le u \tag{9}$$

*Proof.* Let $\mathcal{I} \subseteq \{1, \ldots, c\}$ be the set of all true null hypotheses at $\boldsymbol{\lambda}$. We have:

$$
\begin{aligned}
\mathbb{P}\left(p(\boldsymbol{\lambda}, \boldsymbol{\alpha}) \le u\right) &\le \mathbb{P}\left(\max_{i \in \mathcal{I}} p_i(\boldsymbol{\lambda}, \boldsymbol{\alpha}) \le u\right) \\
&= \mathbb{P}\left(\bigcap_{i \in \mathcal{I}} p_i(\boldsymbol{\lambda}, \alpha_i) \le u\right) \le \max_{i \in \mathcal{I}} \mathbb{P}\left(p(\boldsymbol{\lambda}, \alpha_i) \le u\right).
\end{aligned}
\tag{10}
$$

Since for each $i \in \mathcal{I}$, $\mathbb{P}\left(p_i(\boldsymbol{\lambda}, \alpha_i) \le u\right) \le u$, we have $\max_{i \in \mathcal{I}} \mathbb{P}\left(p(\boldsymbol{\lambda}, \alpha_i) \le u\right) \le u$, which implies $\mathbb{P}\left(p(\boldsymbol{\lambda}, \boldsymbol{\alpha}) \le u\right) \le u$.    □

### A.2    PROOF OF PROPOSITION 5.1

This is direct result of Split FST (Angelopoulos et al., 2021), which we prove here for completeness.

*Proof.* Since $\mathcal{D}_{\text{opt}}$ and $\mathcal{D}_{\text{testing}}$ are disjoint, i.i.d., $\mathcal{D}_{\text{testing}}$ is also i.i.d. w.r.t the returned Pareto frontier over $\mathcal{D}_{\text{opt}}$. Lemma A.1 then gives that $p^{\text{testing}}(\boldsymbol{\lambda}, \boldsymbol{\alpha})$ are super-uniform under $H_{\boldsymbol{\lambda}}$.

We now prove simultaneous $(\alpha, \delta)$-control over $\Lambda_r$. Let $H_{\boldsymbol{\lambda}'}$ be the first true null hypothesis in the sequence. Given that $p(\boldsymbol{\lambda}', \boldsymbol{\alpha})$ is a super uniform p-value under $H_{\boldsymbol{\lambda}'}$, the probability of making a false discovery at $\boldsymbol{\lambda}'$ is bounded by $\delta$. However, if $H_{\boldsymbol{\lambda}'}$ fails to be rejected (no false discovery), then all other $H_{\boldsymbol{\lambda}}$ that follow in the sequence also fail to be rejected (regardless of if $H_{\boldsymbol{\lambda}}$ is true or not). So the probability of making any false discoveries is also bounded by $\delta$.

This implies that the probability that all configurations in $\Lambda^* \subseteq \Lambda_r$ are risk controlling is at least $1 - \delta$, which also implies that any configuration in $\Lambda^*$ is $(\alpha, \delta)$-risk controlling.    □

### A.3    PROOF OF PROPOSITION 5.2

*Proof.* We restate our randomized time-sharing strategy: for each test point $(X, Y)$ we independently sample the configuration $\boldsymbol{\lambda}_j^* \in \Lambda^*$ to use with probability $\Delta_j$, where $\Delta \in \mathcal{S}^{|\Lambda^*|}$ is a point in the $|\Lambda^*| - 1$ dimensional probability simplex $\mathcal{S}^{|\Lambda^*|} = \left\{ \Delta \in \mathbb{R}^{|\Lambda^*|} \big| \sum_j \Delta_j = 1, \Delta_j \ge 1 \right\}$.

Given $\mathcal{D}_{\text{cal}}$, $\Lambda^*$ is a constant set, and each risk $Q_i(\boldsymbol{\lambda}_j^*) = \mathbb{E}[q_i(X, Y; \boldsymbol{\lambda}_j^*)]$ is also a constant for all $\boldsymbol{\lambda}_j^* \in \Lambda^*$ and $i \in \{1, \ldots, c\}$. For each $Q_i$, the combined risk of the time-sharing strategy can then

be derived as the mean of a mixture model where

$$Q_i(\boldsymbol{\lambda}^{\text{share}}) = \sum_{j=1}^{|\Lambda^*|} \Delta_j Q_i(\boldsymbol{\lambda}_j^*)$$

$$\leq \sum_{j=1}^{|\Lambda^*|} \Delta_j \max_{j'} Q_i(\boldsymbol{\lambda}_{j'}^*) = \max_{j'} Q_i(\boldsymbol{\lambda}_{j'}^*).$$

(11)

Let $E$ be the event that all $\boldsymbol{\lambda}_j^* \in \Lambda^*$ are risk controlling across all $Q_i$ at level $\alpha_i$ given the draw of $\mathcal{D}_{\text{cal}}$. Proposition 5.1 gives that this event occurs with probability at least $1 - \delta$. Therefore,

$$\mathbb{P}\left(Q_i(\boldsymbol{\lambda}^{\text{share}}) \leq \alpha_i\right) \geq \mathbb{P}(\max_{j'} Q_i(\boldsymbol{\lambda}_{j'}^*) \leq \alpha_i) \geq 1 - \delta, \ \ \forall i \in \{1, \ldots, c\} \text{ simultaneously.} \quad (12)$$

Thus we have that $\boldsymbol{\lambda}^{\text{share}}$ is also $(\alpha, \delta)$-risk controlling for any choice of $\Delta$. $\qquad\square$

### A.4 HOEFFDING-BENTKUS INEQUALITY P-VALUES

The Hoeffding-Bentkus from (Bates et al., 2021) is a combination of Hoeffding and Bentkus inequalities:

$$p\left(\hat{Q}^{\text{cal}}(\boldsymbol{\lambda}); \alpha, m\right) = \min\left(\exp\{-mh_1(\hat{Q}^{\text{cal}}(\boldsymbol{\lambda}) \wedge \alpha, \alpha)\}, e\mathbb{P}\left(\text{Binom}(m, \alpha) \leq \lceil m\hat{Q}^{\text{cal}}(\boldsymbol{\lambda})\rceil\right)\right)$$

(13)

where $h_1(a, b) = a\log(\frac{a}{b}) + (1-a)\log(\frac{1-a}{1-b})$.

### A.5 WORST-CLASS OBJECTIVE

The empirical risk for the class-conditioned objective $\mathbb{E}\left[D'(X, Y; \boldsymbol{\lambda})|Y = y\right]$ is computed over the samples in class $y$:

$$\hat{Q}_{\text{acc-class}}(y, \boldsymbol{\lambda}) = \frac{1}{|\mathcal{D}_{\text{cal}}^y|} \sum_{(X,Y)\in\mathcal{D}_{\text{cal}}^y} D'(X, Y; \boldsymbol{\lambda}), \ \ \mathcal{D}_{\text{cal}}^y = \{(X, Y) \in \mathcal{D}_{\text{cal}}|Y = y\} \quad (14)$$

Note that the per-class risks are of the same type and are bounded by the same $\alpha$. If furthermore classes are approximately balanced, i.e $|\mathcal{D}_{\text{cal}}^y| \approx |\mathcal{D}_{\text{cal}}^{\tilde{y}}| \ \ \forall y, \tilde{y} \in \mathcal{Y}$, than due to the monotonicity of the p-value with respect to the empirical risk, we have:

$$p(\boldsymbol{\lambda}, \alpha) = \max_{y\in\mathcal{Y}} p\left(\hat{Q}_{\text{acc-class}}(y, \boldsymbol{\lambda}); \alpha, |\mathcal{D}_{\text{cal}}^y|\right) \approx p\left(\max_{y\in\mathcal{Y}} \hat{Q}_{\text{acc-class}}(y, \boldsymbol{\lambda}); \alpha, |\mathcal{D}_{\text{cal}}^y|\right). \quad (15)$$

Therefore, instead of $|\mathcal{Y}|$ objective functions, one per each class, we can define a single equivalent empirical objective $\hat{Q}_{\text{acc-worst}}(\boldsymbol{\lambda}) = \max_{y\in\mathcal{Y}} \hat{Q}_{\text{acc-class}}(y, \boldsymbol{\lambda})$.

### A.6 SELECTIVE CLASSIFICATION

Similarly to above derivation for per-class accuracy, for selective classification we define the selective cost (given selection):

$$Q_{\text{select-cost}}(\boldsymbol{\lambda}, \tau) = \mathbb{E}\left[q_{\text{cost}}(X; \boldsymbol{\lambda}) \Big| \max_y f(X, y; \boldsymbol{\lambda}) \geq \tau\right] \quad (16)$$

and the same for selective accuracy reduction:

$$Q_{\text{select-acc}}(\boldsymbol{\lambda}, \tau) = \mathbb{E}\left[q_{\text{acc}}(X; \boldsymbol{\lambda}) \Big| \max_y f(X, y; \boldsymbol{\lambda}) \geq \tau\right] \quad (17)$$

which are evaluated empirically by:

$$\hat{Q}_{\text{select-cost}}(\boldsymbol{\lambda}, \tau) = \frac{1}{|D_{\text{cal}}^\tau|} \sum_{X\in\mathcal{D}_{\text{cal}}^\tau} q_{\text{cost}}(X; \boldsymbol{\lambda})$$

$$\hat{Q}_{\text{select-acc}}(\boldsymbol{\lambda}, \tau) = \frac{1}{|D_{\text{cal}}^\tau|} \sum_{X\in\mathcal{D}_{\text{cal}}^\tau} q_{\text{acc}}(X; \boldsymbol{\lambda})$$

where $\mathcal{D}_{\text{cal}}^\tau = \{X \in \mathcal{D}_{\text{cal}}| \max_y f(X, y; \boldsymbol{\lambda}) \geq \tau\}$.

## B  MULTI-DIMENSIONAL PRUNING

First, we describe the core model units and introduce some essential notation, while keeping the exact model implementation as general as possible. Second, we describe each of the pruning dimensions, its associated importance score, and the thresholding mechanism.

### B.1  TRANSFORMER MODEL

Consider a Transformer model (Vaswani et al., 2017; Devlin et al., 2018) with $K$ layers. The input to the model is given as a sequence of $L$ (for notational simplicity we omit here the dependency of the length on $x$) tokens $x = (x_1, \ldots, x_L)$, which are first mapped to learneable word embeddings $e = (e_1, \ldots, e_L)$. Tokens are then passed through the model's layers, with $h_j = (h_{j,1}, \ldots, h_{j,L})$ denoting the $j$-th layer hidden representation. Each layer consists of a multi-head attention, with $W$ heads producing the combined output $a_j = \sum_{w=1}^{W} \text{Attn}_{j,w}(h_{j-1})$, which is followed by a feed-forward network to provide the next layer hidden representation. The last layer is attached to a classification head with $|\mathcal{Y}|$ outputs, where $f(x, y)$ denotes the output of class $y$. The model is optimized by minimizing a loss function $\mathcal{L}$ computed empirically over the training set.

### B.2  TOKEN PRUNING

It is often the case that the input consists of a large amount of tokens that have a negligible contribution for the prediction task. The idea in token pruning is to identify unimportant tokens and discard them at some point in the model.

To identify the contribution of each token, we attach to each layer a token importance predictor $s_j^{\text{tok}} : \mathcal{X} \to \mathbb{R}$ based on the token hidden representation $h_{j,l}$. Following (Modarressi et al., 2022), we use as importance scores gradient attributions, computed by:

$$r_l = \left\| \frac{\partial f(x, y_c)}{\partial e_l} \odot e_l \right\|_2 \tag{18}$$

where $y_c$ is the true label, and $\odot$ denotes element-wise product. The token importance predictors in each layer are optimized with a cross-entropy loss, where the labels are the scores, normalised to sum to one. In the $j$-th layer, tokens with $s_j^{\text{tok}}(x_l) < \lambda^{\text{tok}}$ are pruned and are not transferred to next layer. The number of tokens remaining after pruning is given by $L_j(x; \lambda^{\text{tok}}) = \sum_{l=1}^{L} \bigcap_{j'=1}^{j} \mathbf{1} \left\{ s_{j'}^{\text{tok}}(x_l) > \lambda^{\text{tok}} \right\}$.

### B.3  EARLY EXITING

Early-exiting is based on the idea that examples vary in their difficulty level, hence, require different amount of computation to reach to a good prediction. While for simple examples a decision can be made early on, difficult examples may require going through the full model. We attach a prediction head $f_j : \mathcal{X} \to \mathcal{Y}$ to each layer, trained to predict the labels via the same loss function as the original model. Following (Liu et al., 2020), we define the importance score based on the prediction head entropy:

$$s_j^{\text{layer}}(x) = \sum_{y \in |\mathcal{Y}|} p_j(y|x) \log p_j(y|x) \tag{19}$$

where $p_j(y|x)$ are the per-class probabilities provided by the $j$-th prediction head. Based on this score, examples with $s_j^{\text{layer}}(x) < \lambda^{\text{layer}}$ exit in the $j$-th layer. The exit layer of $x$ is given by $K_{\text{exit}}(x; \lambda^{\text{layer}}) = \arg\min_j \left\{ j \in \{1, \ldots, K\} \Big| s_j^{\text{layer}}(x) < \lambda^{\text{layer}} \right\}$.

### B.4  HEAD PRUNING

It was shown by Michel et al. (2019) that a significant fraction of attention heads can be removed with a little impact on the performance. Each attention head $(w, j)$, $1 \le j \le K, 1 \le w \le W$ is assigned a score $s_j^{\text{head}}(w)$ according to:

$$s_j^{\text{head}}(w) = \left| \text{Attn}_{j,w}(h_{j-1})^T \frac{\partial \mathcal{L}}{\partial \text{Attn}_{j,w}(h_{j-1})} \right|. \tag{20}$$

Table C.1: Datasets Details

| Dataset | $|\mathcal{Y}|$ | Task | Train | Val. | Test | Cal. (out of Test) | Full model Acc. [%] |
|---|---|---|---|---|---|---|---|
| IMDB | 2 | Sentiment analysis on movie reviews | 20K | 5K | 10K | 5K | 94 |
| AG News | 4 | News topic classification | 115K | 5K | 7.6K | 5K | 93 |
| QNLI | 2 | Question-answer pair classification | $\sim 10$K | 5K | $\sim 5.5$K | 3.4K | 92 |
| QQP | 2 | Question pair semantic equivalence | $\sim 360$K | 5K | 10K | 5K | 91 |
| MNLI | 3 | Natural language inference | $\sim 250$K | $\sim 150$K | 30K | 10K | 86 |

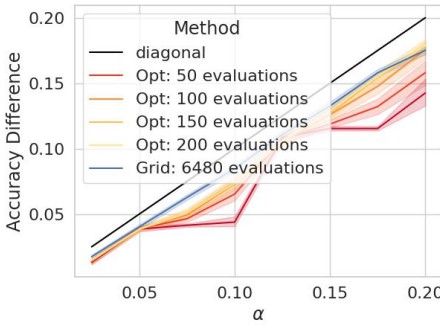 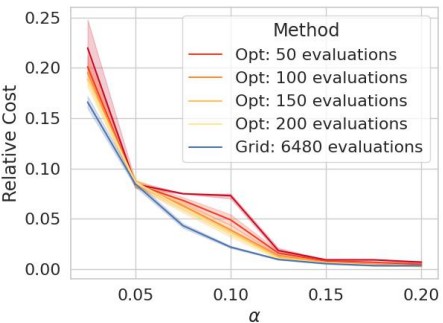

Figure D.1: Results of Pareto Testing over AG News with multi-objective optimizer for different number of evaluations, and with a grid of thresholds. Results are averaged over 50 random splits. Accuracy reduction is controlled and cost is minimized.

The scores in each layer are normalized to sum to one. Attention heads with $s_j^{\text{head}}(w) < \lambda^{\text{head}}$ are pruned. The number of heads left after pruning is given by $W_j(\lambda^{\text{head}}) = \sum_{w=1}^{W} \mathbf{1}\left\{s_j^{\text{head}}(w) > \lambda^{\text{head}}\right\}$. Note that this is a fixed pruning, unlike the previous pruning dimensions that vary according to the input $x$.

## C  IMPLEMENTATION AND DATASET DETAILS

**Datasets.** Splitting specifications and full model performance on each task are contained in Table C.1. Note that for IMDB, QQP and MNLI we used a subset of the original dev/test set in order to expedite evaluation. For MNLI we used the split of (Sagawa et al., 2019).

**Prediction Heads.** Each prediction head is a 2-layer feed-forward neural network with 32 dimensional hidden states, and ReLU activation. The input is the hidden representation of the [CLS] token concatenated with the hidden representation of all previous layers, as was proposed in (Wołczyk et al., 2021).

**Token importance predictors.** Each token importance predictor is a 2-layer feed-forward neural network with 32 dimensional hidden states, and ReLU activation. The input is the hidden representation of each token in the current layer and all previous layers, following (Wołczyk et al., 2021).

**Training.** The core model is first finetuned on each task. We compute the attention head importance scores based on validation data. We freeze the backbone model and train the early-exit classifiers and the token importance predictors on the training data.

**Code.** Our code will be made available at https://github.com/bracha-laufer/pareto-testing.

## D  ADDITIONAL BASELINES AND RESULTS

Our method is based on the Learn then Test (LTT) framework (Angelopoulos et al., 2021), which is summarized in Algorithm F.2. We compare our method to two baselines from (Angelopoulos et al., 2021): 3D-SGT summarised in Algorithm F.3, which is a 3D extension to the 2D Hamming SGT,

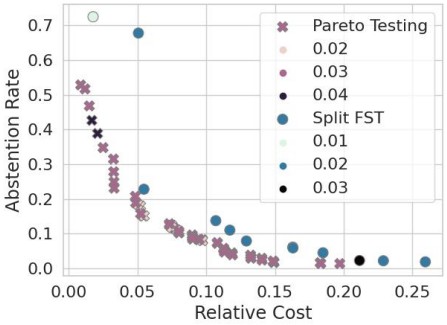

Figure D.2: Three-objectives, AG News - one controlled, two free: accuracy reduction is controlled by $\alpha = 0.05, \delta = 0.1$, cost and abstention rate are minimized. The coloring is according to accuracy reduction.

and Split-FST described in Appendix E. Note that we consider a broader setting in which, besides multiple risk control we wish to optimize additional free objective functions.

In addition, we define two non-risk controlling baselines:

$\alpha$**-constrained** - A constrained optimization problem can be defined by:

$$\min_{\boldsymbol{\lambda} \in \Lambda} \quad \mathbf{q}_k^{\mathrm{cal}}(\boldsymbol{\lambda}) \tag{21}$$
$$\text{s.t.} \quad \hat{Q}_i^{\mathrm{cal}}(\boldsymbol{\lambda}) < \alpha_i, \ \forall 1 \le i \le c,$$

where $\mathbf{q}_k^{\mathrm{cal}}(\boldsymbol{\lambda}) = [\hat{Q}_{c+1}^{\mathrm{cal}}(\boldsymbol{\lambda}), \dots, \hat{Q}_{c+k}^{\mathrm{cal}}(\boldsymbol{\lambda})]^T$. Directly solving for Eq. (21) over the calibration data, however, would not necessarily yield a generalizable $\hat{\boldsymbol{\lambda}}$ with the desired $1 - \delta$ probability. In other words, the true risk $Q_i(\hat{\boldsymbol{\lambda}})$ over test data might exceed $\alpha_i$, possibly with high probability.

$(\alpha, \delta)$**-constrained** - We simultaneously test all possible configurations with error level $\delta$ without correcting for multiple hypothesis testing.

Moreover, we develop two additional baselines and present their results herein:

**Low-Risk Path** - similar to Split-FST (and Pareto Testing), this is a dual-stage method, assuming that the calibration data is split into two subsets. In the first stage, we find a solution to the constrained optimization problem, defined in Eq. (21). Then a low risk path is defined from full model to the solution. The path is defined over the grid of hyper-parameter combinations, where in each step we pick a neighbouring hyper-parameter combination (increasing one hyper-parameter dimension with respect to previous step) with lowest risk among all neighbours. The method is summarized in Algorithm F.4. Note that as the method defines the path in the hyper-parameter space, it implicitly assumes that the objective functions are monotonic with respect to each of the hyper-parameters.

**Constrained-Path Testing** - This can be considered a variant of the proposed method. When interested in a single configuration selection for specific $\boldsymbol{\alpha}$ and $\delta$, a cheaper (but not equivalent) approach would be to solve *multiple* constrained problems:

$$\min_{\boldsymbol{\lambda} \in \Lambda} \quad \mathbf{q}_k^{\mathrm{opt}}(\boldsymbol{\lambda}) \tag{22}$$
$$\text{s.t.} \quad \hat{Q}_i^{\mathrm{opt}}(\boldsymbol{\lambda}) < \alpha_i - \epsilon, \ \forall 1 \le i \le c,$$

for a sequence of $\epsilon$ values in $[0, \dots, \min_i \alpha_i]$. Then, an ordered set of configurations to test is defined by the solutions to Eq. (22) with *decreasing* values of $\epsilon$. Note that both the constrained and the full multi-objective variants are equivalent for the case of a single control constraint. However, when there are multiple constraints Pareto Testing operates on a larger set of hyper-parameter

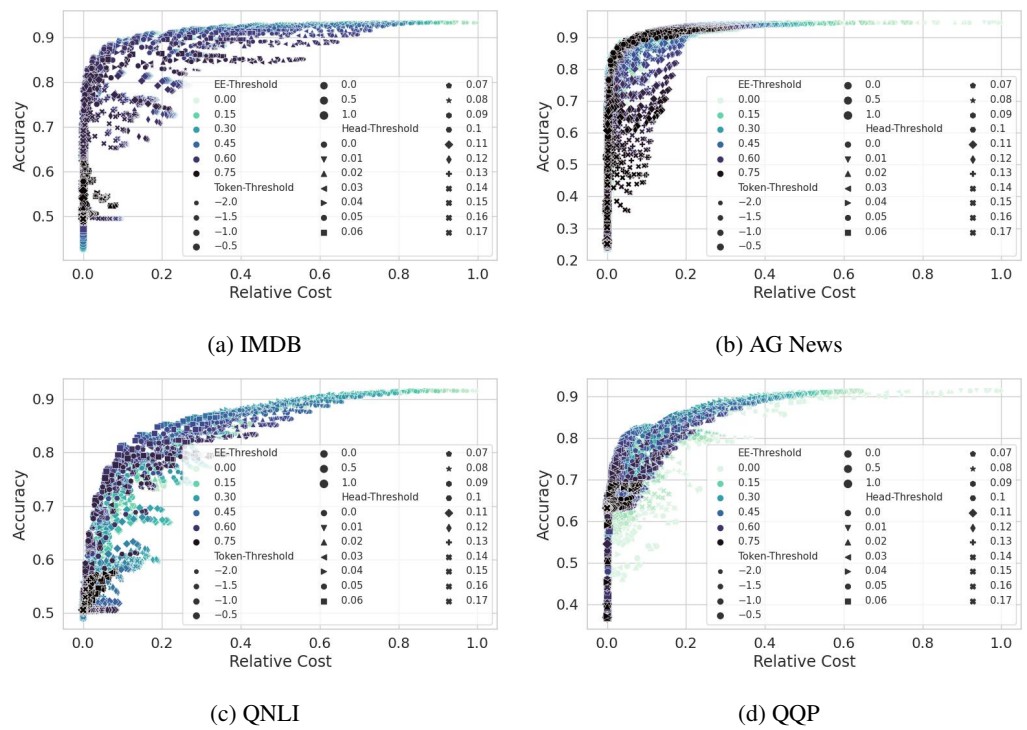

Figure D.3: Shows cost and accuracy trade-offs over test data provided by a grid of $6480$ configurations ($18$ head, $20$ token and $18$ early-exit thresholds).

combinations, consisting of solutions to:

$$\min_{\boldsymbol{\lambda} \in \Lambda} \quad \mathbf{q}_k^{\text{opt}}(\boldsymbol{\lambda}) \tag{23}$$

$$\text{s.t.} \quad \hat{Q}_i^{\text{opt}}(\boldsymbol{\lambda}) < \alpha_i - \epsilon_i, \ \ \forall 1 \leq i \leq c,$$

with $\epsilon_i$ values in $[0, \ldots, \alpha_i]$, namely, solving the constrained problem for all possible combinations of $(\epsilon_1, \ldots, \epsilon_c)$.

**Pruning model.** Figure D.3 shows the accuracy and the relative cost of the proposed adaptive pruning model $f$ for various threshold combinations, computed over test data. We see that the fusion of all pruning dimensions yields a wide variety of configurations with a clear trade-offs between accuracy and cost. In addition, the Pareto front consists of different threshold combinations, indicating that the optimal threshold value in each dimension is not fixed and varies with the desired cost/accuracy level.

**Two objectives - accuracy controlled, cost minimized (additional results).** Results for additional baselines are shown in Fig. D.4, including the two non-controlling baselines: $\alpha$-constrained and $(\alpha, \delta)$-constrained, and our derived Low-Risk Path baseline. We see that, in many cases, Low-Risk Path obtains similar cost reduction compared to our proposed method, however, it is inferior for certain tasks and $\alpha$ values. As expected, both non-risk controlling baselines obtain lower costs compared to our method. This of course comes with a price of suffering from risk violations that exceed $\delta = 0.1$, as can be seen in the bottom line bar plots. However, we see that the there is not a large difference between the cost reductions obtained by our method and the non-controlling baselines. This indicates that our method, though providing risk control guarantees, as opposed to the non-controlling baselines, is not overly conservative, and effectively optimizes the free objective function, leading to significant cost reduction.

We further examine the different aspects of the proposed method. The size of the Pareto optimal set obtained for each task is shown in Fig. D.5. We see that the size of the Pareto set varies among tasks and is around $3 - 5\%$ of the original set ($6480$ configurations for each task), which is a significant

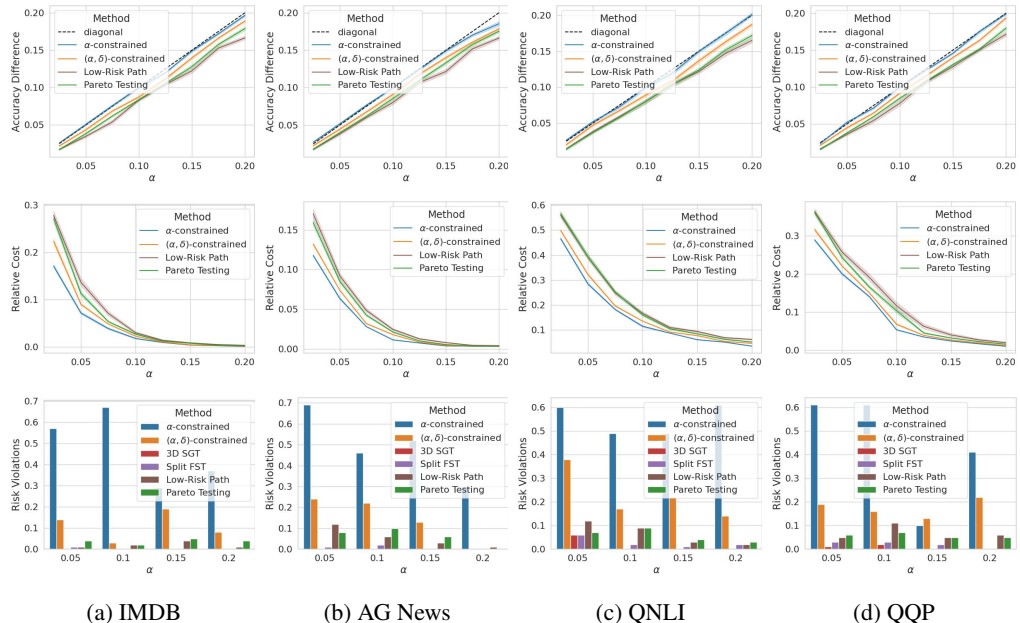

(a) IMDB      (b) AG News      (c) QNLI      (d) QQP

Figure D.4: Additional results for two-objectives - accuracy reduction is controlled by $\alpha \in \{0.025, 0.5, \ldots, 0.2\}$, $\delta = 0.1$, cost is minimized. Top row: accuracy reduction; middle row: relative cost; last row: rate of risk-violations. Results are averaged over 100 random splits to calibration and test.

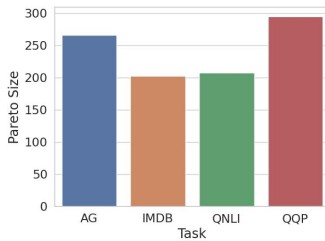

Figure D.5: Additional results for two-objectives: accuracy reduction and cost. Sizes of the Pareto set for each task.

reduction of the hypotheses space. To demonstrate the variation across splits we show the violin plots in Figure D.7. We observe that our method obtains lower variance compared to Spit FST. This may be due to the fact that Split FST orders the hypothesis according to p-values only, and similar p-values can correspond to varying costs. Our method has slightly higher variance compared to 3D SGT, which may be attributed to the additional level of randomization caused by calibration set splitting. Figure D.6 compares the sizes of the validated configurations returned by either Pareto Testing and Split FST, showing that our methods leads to larger sets of valid configurations. In Fig. D.8, we show the results of our method for different sizes of $\mathcal{D}_{\text{opt}}$. We see that as expected the performance improves as the size of $\mathcal{D}_{\text{opt}}$ increases, and that a size of 500 points is already sufficient to obtain stable performance.

**Two objectives - cost controlled, relative accuracy loss minimized**. We evaluated the opposite scenario where the relative cost is controlled, while accuracy reduction is minimized. Note that for 3D-SGT and Low-Risk Path we start testing from the empty model (lowest cost risk) towards the full model. The accuracy reductions of all methods are summarized in Table D.1. We observe that as opposed to accuracy reduction, here the results of all methods are similar (except for Low-Risk Path).

**Two objectives - accuracy controlled, FLOPs minimized.** We experimented with a different cost measure in terms of FLOPs speed-up, which can be considered as a more practical measure, tailored

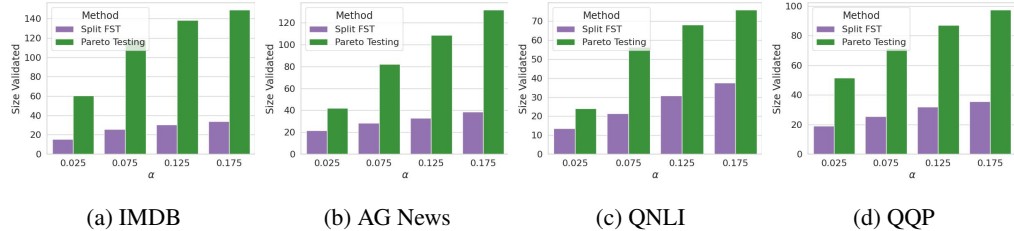

| (a) IMDB | (b) AG News | (c) QNLI | (d) QQP |

Figure D.6: Additional results for two-objectives - accuracy reduction is controlled by $\alpha \in \{0.025, 0.075, 0.125, 0.2\}$, $\delta = 0.1$, cost is minimized. Compare sizes of validated configurations for Pareto Testing and Split FST (computed over 20 random trials).

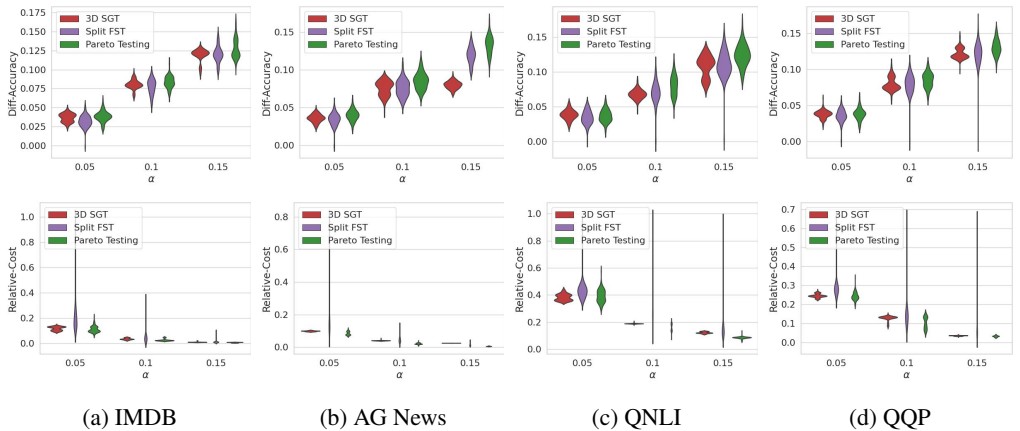

| (a) IMDB | (b) AG News | (c) QNLI | (d) QQP |

Figure D.7: Additional results for two-objectives - accuracy reduction is controlled by $\alpha \in \{0.05, 0.1, 0.15\}$, $\delta = 0.1$, cost is minimized. Top: violin plots for accuracy reduction; bottom: violin plots for relative cost (computed over 100 random splits).

to the specific architecture being used. Results are summarized in Table D.2. We see that the proposed method almost always obtains the best speed-ups, which is inline with our other results.

**Three objectives - accuracy/worst accuracy controlled, cost minimized.** Results for additional baselines are shown in Fig. D.9. Here too, our method performs the best. In this scenario, Constrained-Path Testing obtains similar results while Low-Risk Path is significantly worse.

**Three objectives - accuracy and abstention rate controlled, cost minimized.** We use the same setup of selective classification described in §7. Note that combining $\tau$ with the other three pruning dimensions, we obtain a four dimensional hyper-parameter space, where there is a complex interplay between the hyper-parameters and the risk functions. As $\tau$ increases we expect to get better accuracy-cost trade-offs since we remove difficult examples. In addition, abstention rate is monotonic with respect to $\tau$ but is also influenced by the pruning dimension in an uncharacterized manner.

Here we control both accuracy reduction and abstention rate, while minimizing cost. Since the risk functions are not necessarily monotonic with respect to all hyper-parameters, Low-Risk Path is not relevant. Moreover, since here we have a 4D hyper-parameter space, 3D SGT cannot be applied. Results are summarized in Fig. D.10, where all methods obtain similar results.

## E FAMILY-WISE ERROR RATE CONTROL

Let $\Lambda_g$ denote a set of possible configurations to test, and $\Lambda_r \subseteq \Lambda_g$ denote the set of rejected hypotheses. When performing multiple hypothesis testing (MHT), a FWER-controlling procedure accounts for controlling the probability of making one or more false discoveries, i.e. falsely rejecting at least one true null hypothesis:

$$\mathbb{P}\left(|\Lambda_r \cap \Lambda_0| \geq 1\right) \leq \delta \tag{24}$$

where $\Lambda_0 \subseteq \Lambda_g$ is the set of configurations for which the null hypothesis is true.

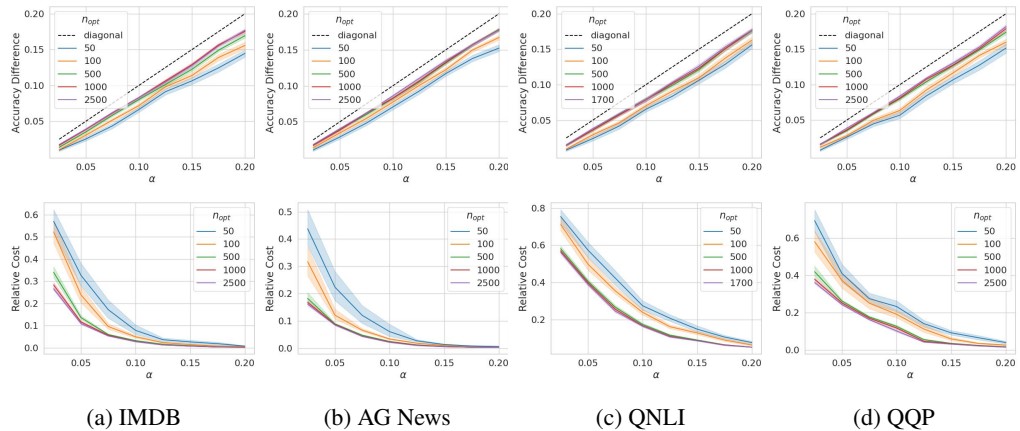

|  | (a) IMDB | (b) AG News | (c) QNLI | (d) QQP |

Figure D.8: Additional results for two-objectives - accuracy reduction is controlled by $\alpha \in \{0.025, 0.05, \ldots, 0.2\}$, $\delta = 0.1$, cost is minimized. Top: accuracy reduction; bottom: relative cost for different sizes of $\mathcal{D}_{\text{opt}}$ (computed over 100 random splits to calibration and test).

Table D.1: Two-objective scenario: relative cost is controlled by $\alpha \in \{0.1, 0.2, 0.3\}$, $\delta = 0.1$, accuracy reduction is minimized. Shows accuracy reduction. Results are averaged over 100 random splits to calibration and test.

| Method | $\alpha = 0.1$ | | | | $\alpha = 0.2$ | | | | $\alpha = 0.3$ | | | |
|---|---|---|---|---|---|---|---|---|---|---|---|---|
| | IMDB | Ag News | QNLI | QQP | IMDB | Ag News | QNLI | QQP | IMDB | Ag News | QNLI | QQP |
| 3D-SGT | 0.05 | 0.04 | 0.14 | 0.09 | 0.03 | 0.02 | 0.09 | 0.06 | 0.02 | 0.00 | 0.05 | 0.03 |
| Split FST | 0.05 | 0.04 | 0.15 | 0.10 | 0.02 | 0.02 | 0.09 | 0.06 | 0.02 | 0.01 | 0.06 | 0.03 |
| Low-Risk Path | 0.26 | 0.10 | 0.22 | 0.25 | 0.17 | 0.05 | 0.26 | 0.11 | 0.16 | 0.01 | 0.12 | 0.05 |
| Pareto Testing | 0.05 | 0.04 | 0.13 | 0.09 | 0.02 | 0.02 | 0.08 | 0.06 | 0.02 | 0.01 | 0.05 | 0.03 |

We briefly describe several possible procedures, some of which exploit a-priori known structure in the hypotheses set.

**Bonferroni Correction.** This is the simplest procedure for counteracting the multiple testing problem, while being also the most conservative. The set of rejected hypotheses retrieved by the Bonferroni correction (Bonferroni, 1936), is given by:

$$\mathcal{H}_{BF} = \left\{ H_{\boldsymbol{\lambda}} : p^{\text{cal}}(\boldsymbol{\lambda}, \boldsymbol{\alpha}) < \delta / |\Lambda_g| \right\} \tag{25}$$

**Fixed Sequence Testing.** Multiplicity correction can be avoided when relying on a *pre-defined* ordering of the hypotheses. In FST, the hypotheses are sequentially tested with the same error budget, until failing to reject for the first time. Denoting by $H_{\boldsymbol{\lambda}^{(1)}}, \ldots, H_{\boldsymbol{\lambda}^{(|\Lambda_g|)}}$ the ordered set of hypotheses, FST yields the following set of rejected hypotheses (Holm, 1979):

$$\mathcal{H}_{FST} = \{H^{(j)} : j < J\}, \;\; J = \min_j \{j : p^{(j)} \geq \delta\}. \tag{26}$$

This procedure is advantageous in the case that there is a natural ordering of the hypotheses from the most likely to be rejected to the least likely one. For example, it can be applied in our problem when $n = 1$ and the hypotheses are ordered by threshold values from low to high.

**Sequential Graphical Testing** SGT (Bretz et al., 2009) can be viewed as an extension to FST, where the relation between the hypotheses is richer than just a sequential path, and is therefore parameterized by a directed graph $G$. The graph's nodes are null hypotheses, and the edges connecting between them specify the way the error budget propagates from one node to the other. Each node is allocated an initial error budget. Each time an hypothesis is rejected, the procedure reallocates the error budget from node $i$ to the rest of the nodes according to the edge's weights, and the graph is modified. Several possible graph structures were proposed in (Angelopoulos et al., 2021) for the case of a two-dimensional grid of hypotheses. One option is a 'Hamming graph' in which the ini-

Table D.2: FLOPs Speed-Up. Accuracy Reduction is controlled by level $\alpha$, while Flops Speed-Up is maximized. Results are averaged over 10 random splits to calibration and test.

| Method | $\alpha = 0.025$ | | | | $\alpha = 0.05$ | | | | $\alpha = 0.1$ | | | |
|---|---|---|---|---|---|---|---|---|---|---|---|---|
| | Ag News | IMDB | QNLI | QQP | Ag News | IMDB | QNLI | QQP | Ag News | IMDB | QNLI | QQP |
| 3D-SGT | ×2.34 | ×1.71 | ×**1.38** | ×**1.89** | ×2.89 | ×4.45 | ×**1.59** | ×2.16 | ×3.12 | ×4.45 | ×1.99 | ×2.71 |
| Split FST | ×2.11 | ×1.69 | ×1.35 | ×1.66 | ×2.38 | ×1.99 | ×1.50 | ×2.15 | ×2.80 | ×4.18 | ×1.89 | ×2.69 |
| Low-Risk Path | ×2.38 | ×3.86 | ×1.37 | ×1.85 | ×2.95 | ×4.2 | ×1.56 | ×2.16 | ×3.65 | ×4.20 | ×1.94 | ×2.73 |
| Pareto Testing | ×**2.42** | ×**4.82** | ×1.37 | ×1.86 | ×**3.00** | ×**4.82** | ×**1.59** | ×**2.18** | ×**3.65** | ×**4.82** | ×**2.02** | ×**2.86** |

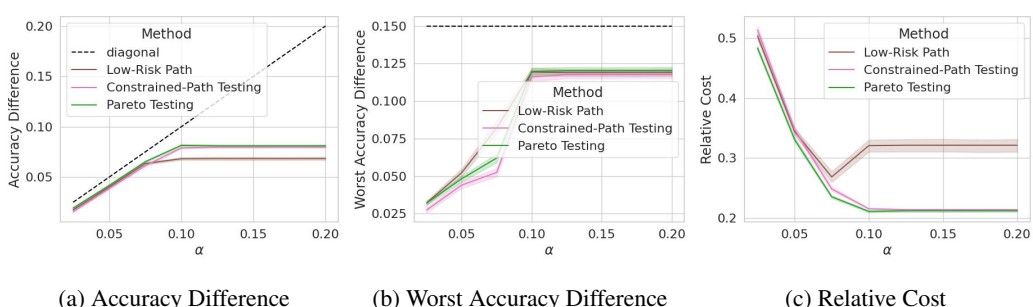

(a) Accuracy Difference  (b) Worst Accuracy Difference  (c) Relative Cost

Figure D.9: Three-objective scenario, MNLI - two controlled, one free: average accuracy is controlled by $\alpha_1 \in \{0.025, 0.5, \ldots, 0.2\}$, worst accuracy is controlled by $\alpha_2 = 0.15$, $\delta = 0.1$, cost is minimized without control. Results are averaged over 100 random splits to calibration and test.

tial error budget is allocated to the bottom-right node, and the error budget is propagated outward. Another option is 'Fallback', where the error budget is split between each possibility in the first dimension. An FST is then performed for a fixed value on the first dimension, and progressing in the other dimension.

**Split Fixed Sequence Testing.** Proposed in (Angelopoulos et al., 2021), Split-FST can be utilized when there is no clear structural relationship between the hypotheses for defining a graph for SGT. The core idea is to split the calibration data in two subsets, where the first split is used to learn the graph, while the other is used for testing. Specifically, they propose to define a sequence of p-values $\beta$ ranging from 0 to 1. Then, for each $\beta$, find the hypothesis where the p-values of all risks (computed over the first split) are the closest (in vector infinity norm) to $\beta$. Based on this ordering, FST can be then performed over the second split.

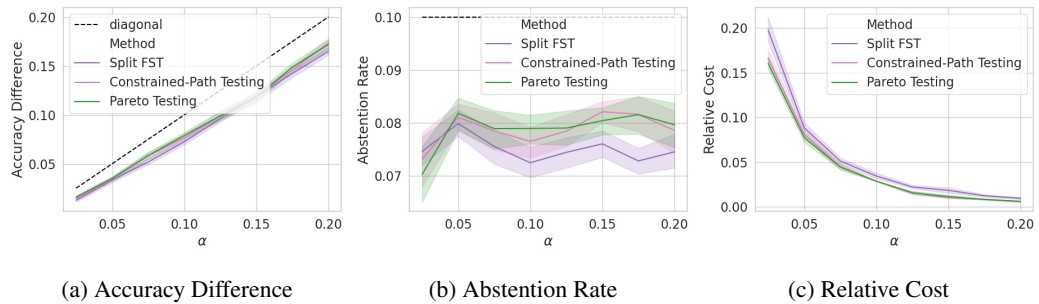

(a) Accuracy Difference        (b) Abstention Rate        (c) Relative Cost

Figure D.10: Three-objective scenario, AG News - two controlled, one free: average accuracy is controlled by $\alpha_1 \in \{0.025, 0.5, \ldots, 0.2\}$, abstention rate is controlled by $\alpha_2 = 0.1$, $\delta = 0.1$, cost is minimized. Results are averaged over 100 random splits to calibration and test.

## F    ALGORITHMS

---

**Algorithm F.1** Recover Pareto Optimal Set

---

**Definitions:** $\Lambda$ is the hyper-parameters search space. $G_1, \ldots, G_r$ is a set of $r$ objective functions. $B$ is the maximum number of function evaluations. `Optimizer` is a multi-objective optimization algorithm.

1: **function** PARETOOPTIMALSET($\Lambda, G_1, \ldots, G_r$)
2:      **if** `Optimizer` = `Brute-Force` **then**
3:          $\Lambda_g \leftarrow$ uniformly sample $B$ configurations in $\Lambda$
4:          $\Lambda_{\text{par}} \leftarrow$ NONDOMINATED($\Lambda_g, G_1, \ldots, G_r$)
5:      **else**
6:          $\Lambda_{\text{par}} \leftarrow$ `Optimizer`($\Lambda, G_1, \ldots, G_r, B$)
       **return** $\Lambda_{\text{par}}$
7: **function** NONDOMINATED($\Lambda_g, G_1, \ldots, G_r$)
8:      $\Lambda_o \leftarrow \Lambda_g$
9:      **for** $\boldsymbol{\lambda} \in \Lambda_g$ **do**
10:          **for** $\boldsymbol{\lambda}' \in \Lambda_g, \boldsymbol{\lambda}' \neq \boldsymbol{\lambda}$ **do**
11:             $d = 0$
12:             **for** $i \in \{1, \ldots, r\}$ **do**
13:                 **if** $G_i(\boldsymbol{\lambda}') \leq G_i(\boldsymbol{\lambda})$ **then**
14:                    $d = d + 1$
15:             **if** $d = r$ **then**
16:                 $\Lambda_o = \Lambda_o / \boldsymbol{\lambda}$
17:                 **break**
       **return** $\Lambda_o$

---

---

**Algorithm F.2** Learn then Test (Single Objective)

---

**Definitions:** configurable model $f$ adapted by $n$ thresholds $\boldsymbol{\lambda} = (\lambda_1, \ldots, \lambda_n)$, calibration data $\mathcal{D}_{\text{cal}}$ of size $m$, objective function $Q$, user-specified control limit $\alpha$ and tolerance level $\delta$, a set of hyper-parameter combinations $\Lambda_g$.

1: **function** CALIBRATION($\mathcal{D}_{\text{cal}}, \Lambda_g, \alpha, \delta$)
2:     **for** $\boldsymbol{\lambda} \in \Lambda_g$ **do**
3:         Associate the null hypothesis:

$$H_{\boldsymbol{\lambda}} : Q(\boldsymbol{\lambda}) \geq \alpha \tag{27}$$

4:         Compute the *empirical* risk over $\mathcal{D}_{\text{cal}}$:

$$\hat{Q}^{\text{cal}}(\boldsymbol{\lambda}) = \frac{1}{m} \sum_{(X,Y) \in \mathcal{D}_{\text{cal}}} q(X, Y; \boldsymbol{\lambda}) \tag{28}$$

5:         Compute Hoeffding p-value (or Hoeffding-Bentkus p-value (13)):

$$p^{\text{cal}}(\boldsymbol{\lambda}, \alpha) = p\left(\hat{Q}^{\text{cal}}(\boldsymbol{\lambda}); \alpha, m\right) = e^{-2m\left(\alpha - \hat{Q}^{\text{cal}}(\boldsymbol{\lambda})\right)_+^2} \tag{29}$$

6:         Recover a subset of thresholds $\Lambda_r \subseteq \Lambda_g$ for which the null hypothesis is rejected, by applying a FWER controlling procedure.
7:     **return** $\Lambda_r$

---

**Algorithm F.3** 3D Graph Testing

---

**Definitions:** configurable model $f$ adapted by $n$ thresholds $\boldsymbol{\lambda} = (\lambda_1, \ldots, \lambda_n)$, calibration data $\mathcal{D}_{\text{cal}}$ of size $m$, objective functions $Q_1, \ldots, Q_c$, user-specified control limits $\boldsymbol{\alpha} = (\alpha_1, \ldots, \alpha_c)$ and tolerance level $\delta$, a grid of $I \times J \times K$ thresholds $\Lambda_g = \{\lambda_1^1, \ldots, \lambda_1^I\} \times \{\lambda_2^1, \ldots, \lambda_2^J\} \times \{\lambda_3^1, \ldots, \lambda_3^K\}$, $\lambda^{i,j,k} = (\lambda_1^i, \lambda_2^j, \lambda_3^k)$, 3D graph $W$ with $I \times J \times K$ nodes and weights $W_{i',j',k' \to i,j,k}$ determining the error propagation, $\mathbf{A}$ an $I \times J \times K$ matrix with initial error budget for each configuration, satisfying $\sum_{i,j,k} A_{i,j,k} = \delta$.

1: **function** CALIBRATE($\mathcal{D}_{\text{cal}}, \boldsymbol{\alpha}, \mathbf{A}, \mathbf{W}$)
2:     **for** $\boldsymbol{\lambda} \in \Lambda_g$ **do**
3:         For $\boldsymbol{\lambda} \in \Lambda_g$ compute $\hat{Q}_i^{\text{cal}}(\boldsymbol{\lambda}) = \frac{1}{m} \sum_{(X,Y) \in \mathcal{D}_{\text{cal}}} q_i(X, Y; \boldsymbol{\lambda})$.
4:         For $\boldsymbol{\lambda} \in \Lambda_g$ compute, compute p-values $p^{\text{cal}}(\boldsymbol{\lambda}, \boldsymbol{\alpha}) = \max_{1 \leq i \leq c} p\left(\hat{Q}_i^{\text{cal}}(\boldsymbol{\lambda}); \alpha_i, m\right)$.
5:     $\Lambda_r \leftarrow$ SGT$\left(W, p^{\text{cal}}(\boldsymbol{\lambda}, \boldsymbol{\alpha})\right)$.
6:     **return** $\Lambda_r$
7: **function** SGT($W, p^{\text{cal}}(\boldsymbol{\lambda}, \boldsymbol{\alpha})$)
8:     $\mathcal{I} = \{1, \ldots, I\} \times \{1, \ldots, J\} \times \{1, \ldots, K\}$
9:     $\Lambda_r \leftarrow \{\}$
10:    $i^*, j^*, k^* = \arg\min_{(i,j,k) \in \mathcal{I}} p^{\text{cal}}(\boldsymbol{\lambda}^{i,j,k}, \boldsymbol{\alpha}) / A_{i,j,k}$
11:    **while** $|\mathcal{I}| \geq 1$ **do**
12:        **if** $p^{\text{cal}}(\boldsymbol{\lambda}^{i,j,k}, \boldsymbol{\alpha}) < A_{i^*,j^*,k^*}$ **then**
13:           $\Lambda_r \leftarrow \Lambda_r \cup \boldsymbol{\lambda}^{i^*,j^*,k^*}$
14:           $\mathcal{I} \leftarrow \mathcal{I} / (i^*, j^*, k^*)$
15:           $A_{i,j,k} \leftarrow A_{i,j,k} + A_{i*,j*,k*} W_{i*,j*,k* \to i,j,k}, \forall (i,j,k) \in \mathcal{I}$
16:        $i^*, j^*, k^* = \arg\min_{(i,j,k) \in \mathcal{I}} p^{\text{cal}}(\boldsymbol{\lambda}^{i,j,k}, \boldsymbol{\alpha}) / A_{i,j,k}$
       **return** $\Lambda_r$

---

---

**Algorithm F.4** Shortest-Path Testing

---

**Definitions:** configurable model $f$ adapted by $n$ thresholds $\boldsymbol{\lambda} = (\lambda_1, \ldots, \lambda_n)$, $\mathcal{D}_{\text{cal}} = \mathcal{D}_{\text{opt}} \cup \mathcal{D}_{\text{testing}}$ is a calibration set of size $m$, split into optimization and (statistical) testing sets of size $m_1$ and $m_2$, respectively, objective functions $Q_1, \ldots, Q_c$, user-specified control limits $\boldsymbol{\alpha} = (\alpha_1, \ldots, \alpha_c)$ and tolerance level $\delta$, hyper-parameter resolution $\boldsymbol{\gamma} = (\gamma_1, \ldots, \gamma_n)$, where $\gamma_j$ is the resolution in the $j$-th dimension, $\boldsymbol{\lambda}_{\text{min}}$ consists of the minimum values of all thresholds, $\boldsymbol{\lambda}_{\text{max}}$ consists of the maximum values of all thresholds.

1: **function** OPTIMIZATION($\mathcal{D}_{\text{opt}}, \boldsymbol{\alpha}$)
2:     Define the constrained problem (21).
3:     Apply constrained optimization to find optimal configuration $\boldsymbol{\lambda}_{\text{opt}}$.
4:     $\Lambda_{\text{opt}} \leftarrow$ CREATEPATH $(\boldsymbol{\lambda}_{\text{min}}, \boldsymbol{\lambda}_{\text{opt}}, \boldsymbol{\gamma}) \cup$ CREATEPATH $(\boldsymbol{\lambda}_{\text{opt}}, \boldsymbol{\lambda}_{\text{max}}, \boldsymbol{\gamma})$.
5:     **return** $\Lambda_{\text{opt}}$
6: **function** CREATEPATH($\boldsymbol{\lambda}^{\text{start}}, \boldsymbol{\lambda}^{\text{end}}, \boldsymbol{\gamma}$)
7:     $\boldsymbol{\lambda}^{(0)} \leftarrow \boldsymbol{\lambda}_{\text{start}}$
8:     $i \leftarrow 1$
9:     **while** $\boldsymbol{\lambda}^{(i)} \neq \boldsymbol{\lambda}^{\text{end}}$ **do**
10:         $p_{\text{min}} \leftarrow \infty$
11:         **for** $j \in \{1, \ldots, n\}$ **do**
12:             $\tilde{\lambda}_j^{(i)} \leftarrow \lambda_j^{(i)} + \gamma_j$
13:             **if** $\tilde{\lambda}_j^{(i)} > \lambda_j^{\text{end}}$ **then**
14:                 **Continue**
15:             **else**
16:                 $\boldsymbol{\lambda}^{\text{next}} \leftarrow \left( \lambda_1^{(i)}, \ldots, \tilde{\lambda}_j^{(i)}, \ldots, \lambda_n^{(i)} \right)$
17:                 **if** $p^{\text{opt}}(\boldsymbol{\lambda}^{\text{next}}, \boldsymbol{\alpha}) < p_{\text{min}}$ **then**
18:                     $\boldsymbol{\lambda}^{(i)} \leftarrow \boldsymbol{\lambda}^{\text{next}}$
19:                     $p_{\text{min}} \leftarrow p^{\text{opt}}(\boldsymbol{\lambda}^{\text{next}}, \boldsymbol{\alpha})$
20:         $i \leftarrow i + 1$
        **return** $\left( \boldsymbol{\lambda}^{(0)}, \ldots, \boldsymbol{\lambda}^{(i)} \right)$
21: **function** CALIBRATION($\mathcal{D}_{\text{testing}}, \Lambda_{\text{opt}}, \boldsymbol{\alpha}, \delta$)         ▷ Same as in Algorithm 1

---

