# OpenReview forum: "Efficiently Controlling Multiple Risks with Pareto Testing"
_ICLR.cc/2023/Conference — ICLR 2023 poster_

### Official Review · Reviewer_xGYM · 2022-10-17

**Confidence:** 2
**Correctness:** 4
**Technical Novelty And Significance:** 3
**Empirical Novelty And Significance:** 2
**Recommendation:** 6

**Clarity, Quality, Novelty And Reproducibility:**

Novel as far as I can tell and the quality seems to be high, but the paper is not very clear.

**Strength And Weaknesses:**

+ important and interesting problem
+ novel angle of analysis

- approach not intuitive to understand
- performance in practice not clear

**Summary Of The Paper:**

The paper proposes a two-stage approach to flexibly optimize and test to find
hyperparameter settings that have provable guarantees with respect to the
optimized objectives. The authors describe their framework and evaluate it
empirically on a Transformer model. The presented results demonstrate the
promise of the proposed method.

**Summary Of The Review:**

The paper studies an interesting and important problem with real-world
consequences. It is well written, although the density of math makes it
challenging to understand. Intuitively, it is difficult to follow the presented
technique -- the paper seems to imply that I can guarantee e.g. 100% accuracy
with 100% probability, which does not make sense in practice. A detailed example
to illustrate this would be helpful, in particular as the risks for the
application are not defined in terms of absolute, but relative performance.

The empirical results are not easy to interpret either as all results are shown
dependent on a specific range of alpha values (chosen how?) and it is unclear
how this affects other approaches that are not controlled. If I decided to use
the proposed method, what results can I expect compared to other hyperparameter
optimizers, e.g. SMAC, when given the same resources?

The authors denote hyperparameters with tau, whereas most of the literature uses
lambda. It would be nice to be consistent with the literature.

In the "Model configuration" section on page 2, the example given for
performance guarantees should be _minimal_ error rates. The labels in most
figures are too small.

---

> ### Author Response · Authors · 2022-11-13
> **Response**
>
> Thank you for your helpful and constructive review.
>
> > On the presentation
>
> In our updated manuscript we have significantly revised portions of the paper to be clearer and more intuitive. While one *can* specify 100% accuracy with 100% probability, of course this may not be realizable, and our method will return an empty validated set. We explain the possibility of infeasible constraints in the manuscript (see Footnote 1, p.2). A more typical case would be to request the error increase of the pruned model would be at most  5%  with probability of at least 90%, while also minimizing inference cost. To clarify, we added more examples in the fourth paragraph in the introduction.
>
> > On the empirical results
>
> We focused our evaluation on relative measures as the application considers adaptively pruning computations in Transformer models, while being cautious on the performance change caused by such reductions. We therefore demonstrate a range of values of  $\alpha$ that is practically relevant to our problems of interest. For example, bounding the reduction in accuracy after pruning by a reasonable percentage between 2.5%-20%.
>
> Note that the contribution of our method is not a new hyper-parameter optimizer, but rather a framework under which to combine multi-objective optimization (using standard solvers as smac) with rigorous risk control. Without this validation step, the risk over the test data may be quite large (we show this for our non-risk controlling baselines in Figure. 4).
>
> > On notational consistency
>
> We have changed our use of $\tau$ to $\lambda$ for consistency.
>
> > On minimal error rates
>
> Here we were actually referring to guaranteeing upper bounds on the error rate (as part of risk control). We have updated the confusing phrasing.
>
> > On Figures
>
> We have enlarged the font on all figures.

---

> > ### Comment · Reviewer_xGYM · 2022-11-28
> > **Thank you**
> >
> > Thank you for the update. I have had a look at the new version of the paper, and while it is improved, I'm still not entirely clear on some points.

---

> > > ### Author Response · Authors · 2022-11-28
> > > **Thank you for your response**
> > >
> > > Thank you for reading our response and the changes to our manuscript. We are glad that you found it improved (and would appreciate an increase in score if we have cleared up any weaknesses you saw). We would also be happy to clarify any further points that remain unclear.

---

### Official Review · Reviewer_R3DQ · 2022-10-27

**Confidence:** 5
**Clarity, Quality, Novelty And Reproducibility:** See above.
**Correctness:** 4
**Technical Novelty And Significance:** 3
**Empirical Novelty And Significance:** 3
**Recommendation:** 8

**Strength And Weaknesses:**

Strengths:
- Writing, while very dense, is generally well structured and includes the main points to follow the paper despite building on extensive work in the Learn Then Test paper.
- Related work is covered thoroughly and differentiates the contributions of the paper well.
- The method generally addresses an important problem and improves over previous work by decreasing the number of hypotheses to test.

Weaknesses:
Some comments on writing:
-- Because the paper is very dense, it would occasionally be worth resorting to easier sentences. The fourth paragraph in the introduction is a good example for this.
-- I found the intro figure a bit misleading as it puts too much focus on the architecture. I understand that the experiments use architectural hyper-parameters for risk control. But the figure does not really tell me anything about the risk control itself. It is not a bad figure, but I feel it misses the main contributions.
-- The notation of minimizing a set of is a bit unclear – I assume the authors want to emphasize that we jointly want to minimize different risks. But I feel this should be made clearer – see Eq. (4) and (6) and algorithm 1.
-- The computation of the p-values – which is an integral part of the method – could be made clearer in the main paper. I could only find that in algorithm F.1 or related work and I feel the paper would benefit from making that clearer for readers less familiar with these approaches (e.g., conformal p-values are computed quite differently compared to p-values based on these bounds).

Method:
- How exactly is the Pareto frontier found in practice. The experiments include one paragraph on this, but it is actually unclear what the default method of optimization is? Is it right that exhaustive optimization over the grid is used?

Experiments:
- In general, I feel that the main contribution is improving statistical efficiency and reducing the number of hypotheses to test. I would appreciate some more concrete experiments surrounding this:
-- More explicitly showcases to what extent Pareto Testing reduces the hypotheses space – i.e., how big is the Pareto optimal set, how many hypotheses are tested compared to LTT?
-- How does the D_opt set size impact that and influence the variation or the tightness of the risk control in the end?
- In plots, I assume the are is the standard deviation, or are the min/max/X% confidence interval? I am trying to understand whether Pareto Testing has some influence on this variation.

**Summary Of The Paper:**

The authors propose Pareto Testing, a multiple hypothesis testing strategy based on Learn then test (LTT) for statistically controlling multiple risks. In contrast to prior work, the method uses an additional split of the data to identify hyper-parameter configurations that are Pareto optimal (i.e., on the Pareto frontier). This improves statistical efficiency as less hypothesis tests are required.

**Summary Of The Review:**

I think the paper is a good contribution for ICLR. Due to the density, I feel that writing can be improved at places and I’d appreciate comments on my other questions regarding experiments.

---

> ### Author Response · Authors · 2022-11-13
> **Response**
>
> Thank you for your helpful and constructive review.
>
> > On writing
>
> Thank you for the suggestions. We have uploaded a new revision with significantly improved clarity (e.g., paragraph four of the introduction). We have highlighted sections with major edits in magenta.
>
> > On Figure 1
>
> Thanks for pointing this out. We have updated Figure 1. The new version emphasizes the connection between how our risk control algorithm works, and its impact on our motivating application to Transformers.
>
> > On notation
>
> In our uploaded revision we have introduced more rigorous notation for this kind of multi-objective optimization problem. We have also more clearly denoted that the solution is a set of Pareto optimal points.
>
> > On p-value computation
>
> Following your suggestion, we added more discussion on how p-values are computed. In the main text we use the more intuitive Hoeffding-based p-values for simplicity, but still include Hoeffding-Bentkus p-values (which we ultimately use) in Appendix A.
>
> > On how the Pareto frontier is found in practice
>
> For fair comparison across algorithms, in all of our main experiments we simply define a grid of 6480 total possible configurations. To find the Pareto frontier out of this grid, we employ exhaustive search (by enumerating all configurations, and keeping only those that are non-dominated).
> In general, however, we can also use (far more efficient) multi-objective hyper-parameter optimizers to find the frontier, such as ParEGO (with the implementation provided as part of the SMAC3 library). Figure D.1 compares the performance of ParEGO (over the full configuration space, not discretized) as a function of its provided budget (i.e., number of iterations) to the brute force solution that we defined over the 6480 grid. As discussed in the second to last paragraph of the results section, even with a relatively small budget (e.g., 50 iterations) we can obtain strong performance.
>
> > On additional experiments
>
> As suggested, we have added additional results to the Appendix (see Figures D.5-D.8). Figure D.5 shows how large the found Pareto set is across tasks (again, here using the exhaustive search approach over the 6480 sized grid). The size of this set is ~200-300 configs depending on the task (i.e., <5% of the total grid space). Figure D.6 compares the number of rejected hypotheses (valid configs found) by both Split FST and Pareto Testing across different $\alpha$. In general, we find that Pareto Testing is able to reject far more hypotheses (which, as expected, is then reflected by the better empirical results). Finally, Figure D.8 plots the effect of the optimization set size, showing that with a set size of 500 points we already get good performance.
>
> > On variation in plots
>
> The plots show 95% CIs for the mean over trials computed via bootstrapping (we have clarified these details in the revised paper). We have also added violin plots in Figure D.7 to demonstrate variation across splits. Pareto Testing yields lower variance than Split FST, while having slightly more variance than SGT.

---

### Official Review · Reviewer_ZXwJ · 2022-11-02

**Confidence:** 3
**Clarity, Quality, Novelty And Reproducibility:** This paper is well-written, and the f…
**Correctness:** 3
**Technical Novelty And Significance:** 2
**Empirical Novelty And Significance:** 3
**Recommendation:** 6

**Strength And Weaknesses:**

Strength:

This work is a great attempt by combining the combination of multi-objective optimization and "Learn and Test" for risk-control.  Intuitively, it is obvious that applying FST only on Pareto Front will be efficient (because the size of configuration is much smaller).

Weaknesses:

The setting discussed in this paper is as:  for all c+k objectives,  the first c objectives are controlled by c user-specified risk bounds, and the rest k objectives are free to optimize.  If k =0, i.e., all the risks are controlled, how to decide the final configuration?

Other minor comments:

(1) Equation(3),  minimizing the remaining objective function $Q_c$? ( I guess it should be $Q_{c+1}$)

(2) Algorithm 1 line 11, when k>1, how to get a set of configurations representing different possible trade-offs?

(3) Figure 2 is a bit confusing, from the path showed, Q1 is not controlled under $\alpha$, instead, the configuration selected with $Q_1>\alpha$.

**Summary Of The Paper:**

This work is built upon the results of Learn Then Test framework by introducing Pareto Testing.  The authors develop an efficient method for calibrating models such that their predictions provably satisfy multiple explicit and simultaneous statistical guarantees (e.g., uppper-bound error rates), while also optimizing any number of additional, unconstrained objectives.

This method contains two stages. Stage1: solving an unstrained  multi-objective optimization problem to recover an approximate set of Pareto-optimal configurations; Stage2: performing rigorous sequential testing over the recovered set to yield tight control of the desired risks.

The main idea of this proposed method is that  to perform testing ONLY over the Pareto optimal set. And the effectiveness is demonstrated for reliably accelerating Transformers


**Summary Of The Review:**

Overall, this paper is marginally above the acceptance threshold.

---

> ### Author Response · Authors · 2022-11-13
> **Response**
>
> Thank you for your helpful and constructive review.
>
> > On the setting
>
> We implicitly assume in our setting that k > 0, i.e., that there are always objectives related to prediction “quality” that one wants to optimize. An example is conformal prediction, where one wishes to maintain a minimum prediction coverage, but also have small set sizes.
>
> Note that we can also have constraints on these k “free” objectives by simply repeating them in the first c, e.g., $\{Q_1, \ldots, Q_r\} = \{Q_{c + 1}, \ldots, Q_{c + r}\}$ if $r$ out of the $k$ optimized objectives also have constraints. Practically, this would have the consequence of declining to return any configuration unless the best found values of these optimized objectives are also below the prescribed limits $\alpha_i$. For example, we may want to find the configuration with the lowest error rate that is also constrained to be at most 10%. If the best solutions on the Pareto frontier with respect to error rates cannot be validated for error rates less than 10%, then we will fail to return a solution.
>
> For the purposes of this paper, we chose to focus our exposition on the natural case where k > 0 and $Q_{1:c}$ and $Q_{c+1:c+k}$ are disjoint.
>
> > On other minor comments
>
> Thanks for pointing these out. We have fixed typos, used better notation in Algorithm 1, and corrected Figure 2 (it was indeed backwards).

---

### Official Review · Reviewer_F3Rn · 2022-11-03

**Confidence:** 4
**Clarity, Quality, Novelty And Reproducibility:** Not clear enough. See comments above.
**Correctness:** 3
**Technical Novelty And Significance:** 2
**Empirical Novelty And Significance:** 3
**Recommendation:** 5

**Strength And Weaknesses:**

Strength. Reducing high-dim configurations is an interesting topic. Experiments are comprehensive.

Weakness:
1. Writing is not clear.  For example, is equation 3, should the optimization objective be of subscript c+1 instead of c? The discussion of c+1 and c+k is not clear enough, constantly switching between the two cases.
2. Lack of innovation. The main technique is based on FST. The only contribution is to suggest using the Pareto frontier to reduce the dimensionality of (tau_1, \cdots, tau_n)'s need to be considered, which is quite natural. Also, there is a discrepancy between the empirical Pareto frontier based on Q.hat and based on the real Q. The one based on real Q is the desired frontier.
3. Notation sloppiness. In equation 4, argmin is mis-used, which is also pointed out by the author themsevle -- it is possible that no uniformly optimal solution exists.

I suggest the authors significantly polish this writing and consider more extensions.

**Summary Of The Paper:**

This paper considers combining multi-objective optimization with multiple-hypothesis testing. More specifically, they consider optimizing some objectives Q_i's while applying risk control on multiple other objectives such that Q_j\le c with high probability. They consider reducing the dimension of the high dimensional vector of candidates of configurations by only restricting themselves to the Pareto frontier. Effectiveness of their method is considered on large-scale transformer models in NLP.

Main method is based on existing work fixed sequential testing, by adapted it to the Pareto frontier setting.



**Summary Of The Review:**

See comments above.

---

> ### Author Response · Authors · 2022-11-13
> **Response**
>
> Thank you for your helpful and constructive review.
>
> > On clarity of the writing
>
> Thank you for pointing this out. We have uploaded a new revision of the manuscript that is significantly clearer (based also on review comments), and with typos corrected.
>
> > On innovation
>
> We would like to clarify that while we build on well-established, rigorous testing tools, the primary contribution of our paper is algorithmic and practical. Searching through combinatorially large configuration spaces in adaptive computation is non-trivial — especially if we wish to also provide rigorous risk control — and can easily result in poor computational and/or statistical efficiency. The proposed method of structuring this search in terms of ordered Pareto frontier sets (Algorithm 1) is indeed natural and effective and results in guaranteed risk control. Empirically, we show that our method enables us to efficiently control several risks simultaneously by configuring multiple “knobs” (e.g., token pruning, early exiting, head pruning) in a large adaptive language model based on Transformer architecture.
>
>
> > On $\hat{Q}$ vs $Q$
>
> It is true that we use the empirical Pareto frontier, and that there may be a discrepancy with the true (desired) Pareto frontier. Part of the challenge of our problem is exactly that we do not know the true value of $Q_i$, and can only estimate it, or bound it with high probability (i.e., this is the risk control element of our method).
>
> Our optimization stage is therefore (necessarily) only an approximation—that we then rigorously test in the second stage in order to return the validated set of configurations that we guarantee to be risk controlling with the desired probability.
>
> > On notation
>
> In our uploaded revision we have introduced more precise notation for this kind of multi-objective optimization problem. We have also more clearly denoted that the solution is a set of Pareto optimal points.
>
> > On extensions
>
> We have added a few more experiments—see also our response to Reviewer R3DQ.

---

> > ### Comment · Reviewer_F3Rn · 2022-12-04
> > **Thanks for the response**
> >
> > Thanks for the response.
> >
> > Regarding Q.hat, seems this step is completely heuristic instead of considering the randomness of Q.hat along with other uncertainty, is that right?
> >
> > I raise my score for other clarification.

---

> > > ### Author Response · Authors · 2022-12-05
> > > **Thanks for the suggestion**
> > >
> > > Thank you for reviewing our updated manuscript and for your response. Yes, the reviewer is correct that defining the initial Pareto frontier based on empirical risks over the optimization data split is a heuristic. It’s an interesting suggestion to also consider uncertainty in this phase---one can imagine that we can identify and use more/less conservative Pareto frontiers based on our confidence in certain empirical risks. We didn’t consider that in this work, but it could be interesting to see if it may provide benefits in future follow-ups! Still, we do emphasize that these approaches only may increase or decrease the ultimate efficiency of the testing stage (which is not heuristic, and does have formal guarantees). Guaranteeing that we recover the true Pareto set is not necessary for risk control.

---

### Author Response · Authors · 2022-11-13
**General response to all reviewers**

We appreciate the thoughtful and constructive comments of all of the reviewers. We have uploaded a significantly revised version of our paper which incorporates the review comments. We believe that the new version is significantly clearer, based on the helpful suggestions. Please let us know of any remaining questions or concerns!

---

### Author Response · Authors · 2022-12-02
**Thank you to all reviewers**

Dear reviewers,

Once again, thank you for your time and efforts in reviewing our paper, and for your valuable feedback.
We would greatly appreciate it if you could go over our response and revised manuscript, and please let us know if they resolve your questions, or if there are any further concerns that need to be addressed.

---

### Decision · Program_Chairs · 2023-01-20

**Decision:**

Accept: poster

**Justification For Why Not Higher Score:**

The paper had a very good rating (6.25) with an average confidence of 3.5. The paper was considered borderline as its grade is in the margin of 4.5 and 6.25. This being said, the paper has practical benefits for the community.

**Justification For Why Not Lower Score:**

Good submission with practical benefit to the community.

**Metareview: Summary, Strengths And Weaknesses:**

I Summary

- I.1 Investigated Problem:
To overcome issues caused by the changing nature of ML frameworks objectives as well as the unreliability and sub-optimality of their corresponding solutions, the paper proposes Pareto Testing.

- I.2 Proposed Solution:
The presented solution is a multiple-hypothesis testing strategy based on the Learn Then Test (LTT) framework for statistically controlling multiple risks. The presented solution calibrates models such that their predictions are proved to satisfy multiple and simultaneous statistical guarantees while it optimizes an arbitrary number of auxiliary objectives. Pareto testing is a two-stage process where it constructs at first an approximate set of Pareto-optimal configurations which is recovered by solving an unconstrained multi-objective optimization problem. Then, rigorous sequential testing over the recovered set, which yields tight control of the desired risks, is performed.

- I.3 Validity Proof of the Proposed Solution:
Empirical evidence is provided to support the validity of the proposed solution. The authors demonstrate the computational effectiveness of the method as it accelerates the execution of large-scale Transformer models in natural language processing (NLP) applications.

II Strengths

- II.1 From a structural point of view: The paper is organized and well-structured. It is important to mention that prior to the rebuttal, the reviewers raised concerns about the dense aspect of the submission despite its well-structured nature. The authors addressed positively this issue in the rebuttal as well as issues related to the clarity and consistency of the notation.

- II.2 From an analytical point of view:  Reviewers pointed out the interesting nature of the problem investigated by the authors in this paper and its practical benefit. It offers a new angle of analysis using Pareto testing which reduces the number of hypotheses tested and therefore provides statistical efficiency improvement. More precisely. Reviewers appreciated:
 . The interesting nature of the addressed problem and improvement upon existing work;
 . The Learn Then Test (LTT) framework is leveraged in a meaningful way.
 . The proposed method has the potential to be of great benefit to the ML community from a practical perspective.

- II.3 From a perspective of soundness (development, unity, and coherence) and completeness (correctness): The strength points mentioned above are sufficient evidence of the soundness and completeness of the paper. The authors build their solution upon existing work. The investigated problem was already faced by one of the reviewers who enjoyed reading the submission. This reinforces the soundness aspect of the presented solution.

An additional point reinforcing the strengths mentioned above is the active interaction of the authors during the rebuttal period and their openness to concerns and questions raised by the reviewers.

III What can be thought of as weaknesses:

- III.1 All reviewers during the AC-reviewer meeting pointed out that the work is less innovative from a theoretical perspective given that the paper is presented from a theoretical background. As a theory paper, more contribution is expected from this side. This does not diminish the practical potential of the proposed solution and its benefit to the community of machine learning.

- III.2 One of the reviewers mentioned during the AC-reviewer meeting that discrepancy in sample complexity is not taken into account.

Most of the points that could be thought of as weaknesses have been addressed.

IV. Potential of the paper:

- IV.1 From a Potential perspective (Potential of the paper to the community): Application-wise, and as it has already been underlined in the strengths section, the proposed solution has the potential to be of great benefit to the ML community. Once the decision of acceptance is announced, providing the source code could be of great benefit to the researchers interested in multi-objective and Pareto settings. We thank the authors for providing a detailed pseudo-code in section F of the paper. Providing source code would make the submission even more impactful in the long term and provide it more openness and accessibility.


**Note From Pc:**

if the above contains the word "oral" or "spotlight" please see: "oral" presentation means -> notable-top-5% and "spotlight" means -> notable-top-25%. As stated in our emails, we are disassociating presentation type from AC recommendations

**Summary Of Ac-Reviewer Meeting:**

As mentioned in the Strengths And Weaknesses sections, all reviewers appreciated the practical benefit of the submission and expressed the fact that it lacks contributions from a theoretical point of view given that it is presented in a theoretical way. Again, we emphasize that this does not diminish the practical potential of the proposed solution and its benefit to the community of machine learning. Unanimously, the reviewers leaned toward the acceptance of the submission during the AC-reviewer meeting.